# Wasserstein Distance Rivals Kullback-Leibler Divergence for Knowledge Distillation

**Jiaming Lv**[*]**, Haoyuan Yang**[*]**, Peihua Li**[†]

Dalian University of Technology

ljm_vlg@mail.dlut.edu.cn, yanghaoyuan@mail.dlut.edu.cn, peihuali@dlut.edu.cn

## Abstract

Since pioneering work of Hinton et al., knowledge distillation based on Kullback-Leibler Divergence (KL-Div) has been predominant, and recently its variants have achieved compelling performance. However, KL-Div only compares probabilities of the corresponding category between the teacher and student while lacking a mechanism for cross-category comparison. Besides, KL-Div is problematic when applied to intermediate layers, as it cannot handle non-overlapping distributions and is unaware of geometry of the underlying manifold. To address these downsides, we propose a methodology of Wasserstein Distance (WD) based knowledge distillation. Specifically, we propose a logit distillation method called WKD-L based on discrete WD, which performs cross-category comparison of probabilities and thus can explicitly leverage rich interrelations among categories. Moreover, we introduce a feature distillation method called WKD-F, which uses a parametric method for modeling feature distributions and adopts continuous WD for transferring knowledge from intermediate layers. Comprehensive evaluations on image classification and object detection have shown (1) for *logit distillation* WKD-L outperforms very strong KL-Div variants; (2) for *feature distillation* WKD-F is superior to the KL-Div counterparts and state-of-the-art competitors. The source code is available at http://peihuali.org/WKD.

## 1 Introduction

Knowledge distillation (KD) aims to transfer knowledge from a high-performance teacher model with large capacity to a lightweight student model. In the past years, it has attracted ever increasing interest and made great advance in deep learning, enjoying widespread applications in visual recognition and object detection, among others [1]. In their pioneering work, Hinton et al. [2] introduce Kullback-Leibler divergence (KL-Div) for knowledge distillation, where the prediction of category probabilities of the student is constrained to be similar to that of the teacher. Since then, KL-Div has been predominant in logit distillation and recently its variants [3; 4; 5] have achieved compelling performance. In addition, such logit distillation methods are complementary to many state-of-the-art methods that transfer knowledge from intermediate layers [6; 7; 8].

Despite the great success, KL-Div has two downsides that hinder fully transferring of the teacher's knowledge. First, KL-Div only compares the probabilities of the corresponding category between the teacher and student, lacking a mechanism to perform cross-category comparison. However, real-world categories exhibit varying degrees of visual resemblance, e.g., mammal species like dog and wolf look more similar to each other while visually very distinct from artifact such as car and bicycle. Deep neural networks (DNNs) can distinguish thousands of categories [9] and thus are

---

[*] These authors contributed equally to this work and share first authorship. [†] Corresponding author. The work was supported by National Natural Science Foundation of China (62471083, 61971086).

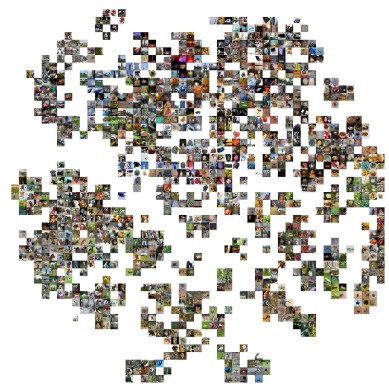

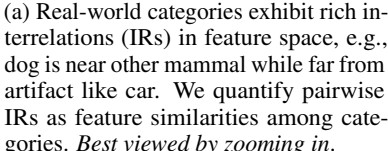

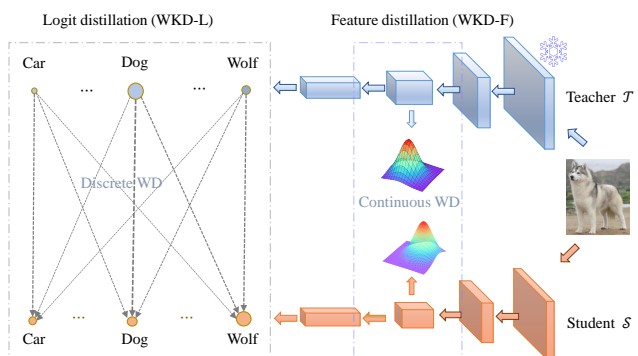

(a) Real-world categories exhibit rich interrelations (IRs) in feature space, e.g., dog is near other mammal while far from artifact like car. We quantify pairwise IRs as feature similarities among categories. *Best viewed by zooming in*.

(b) For logit distillation, discrete WD performs cross-category comparison by exploiting pairwise IRs, in contrast to KL-Div that is a category-to-category measure and lacks a mechanism to use such IRs (cf. Figure 2). For feature distillation, we use Gaussians for distribution modeling and continuous WD for knowledge transfer.

Figure 1: Our methodology of Wasserstein Distance (WD) based knowledge distillation. To effectively exploit rich category interrelations (a), we propose discrete WD based logit distillation (WKD-L) (b) that matches predicted distributions between the teacher and student. Besides, we introduce a feature distillation method based on continuous WD (WKD-F) (b), where we let student mimic parametric feature distributions of the teacher. In (a), features of 100 categories are displayed by the corresponding images as per their 2D embeddings obtained by t-SNE; *refer to Section A.1 for details on this visualization*.

well-informed of such complex relations among categories, as shown in Figure 1a. Unfortunately, due to its category-to-category nature, the classical KD [2] and its variants [3; 4; 5] are unable to explicitly utilize this rich cross-category knowledge.

Secondly, KL-Div is problematic for distilling knowledge from intermediate layers. Deep features of an image are generally high-dimensional and of small size, so being populated very sparsely in the feature space [10, Chap. 2]. This not only makes non-parametric density estimation (e.g., histogram) that KL-Div requires infeasible due to curse of dimensionality, but also leads to non-overlapping discrete distributions that KL-Div fails to deal with [11]. One may turn to parametric, continuous methods (e.g., Gaussian) for modeling feature distributions. However, KL-Div and its variants have limited ability for measuring dis-similarity between continuous distributions, as it is not a metric [12] and is unaware of geometric structure of the underlying manifold [13].

The Wasserstein distance (WD) [14], also called Earth Mover's Distance (EMD) or optimal transport, has the potential to address the limitations of KL-Div. The WD between two probability distributions is generally defined as the minimal cost to transform one distribution to the other. Several works have made exploration by using WD for knowledge transfer from intermediate layers [15; 16]. Specifically, they measure the dis-similarity of a mini-batch of images between the teacher and student based on discrete WD, which concerns comparison across instances in a soft manner, failing to make use of relations across categories. Moreover, they mainly quest for non-parametric method for modeling distributions, behind in performance state-of-the-art KL-Div based methods.

To address these problems, we propose a methodology of Wasserstein distance based knowledge distillation, which we call WKD. This methodology is applicable to logits (WKD-L) as well as to intermediate layers (WKD-F) as shown in Figure 1b. In WKD-L, we minimize the discrepancy between the predicted probabilities of the teacher and student using discrete WD for knowledge transfer. In this way, we perform cross-category comparison that effectively leverages interrelations (IRs) among categories, in stark contrast to category-to-category comparison in the classical KL-Div. We propose to use Centered Kernel Alignment (CKA) [17; 18] to quantify category IRs, which measures the similarity of features between any pair of categories.

For WKD-F, we introduce WD into intermediate layers to condense knowledge from features. Unlike the logits, there is no class probability involved in the intermediate layers. Therefore, we let the student directly match the feature distributions of the teacher. As the dimensions of DNN features are

high, the non-parametric methods (e.g., histogram) are infeasible due to curse of dimensionality [10, Chap. 2], we choose parametric methods for modeling distributions. Specifically, we utilize one of the most widely used continuous distributions (i.e., Gaussian), which is of maximal entropy given 1st- and 2nd-moments estimated from features [19, Chap. 1]. WD between Gaussians can be computed in closed form and is a Riemannian metric on the underlying manifold [20].

We summarize our contributions in the following.

- We present a discrete WD based logit distillation method (WKD-L). It can leverage rich interrelations among classes via cross-category comparisons between predicted probabilities of the teacher and student, overcoming the downside of category-to-category KL divergence.
- We introduce continuous WD into intermediate layers for feature distillation (WKD-F). It can effectively leverage geometric structure of the Riemannian space of Gaussians, better than geometry-unaware KL-divergence.
- On both image classification and object detection tasks, WKD-L perform better than very strong KL-Div based logit distillation methods, while WKD-F is supervisor to the KL-Div counterparts and competitors of feature distillation. Their combination further improves the performance.

## 2  WD for Knowledge Transfer

Given a pre-trained, high-performance teacher model $\mathcal{T}$, our task is to train a lightweight student model $\mathcal{S}$ that can distill knowledge from the teacher. As such, supervisions of the student are from both the ground truth label with the cross entropy loss and from the teacher with distillation losses to be described in the next two sections.

### 2.1  Discrete WD for Logit Distillation

**Interrelations (IRs) among categories.**  As shown in Figures 1a and 4, real-world categories exhibit complex topological relations in the feature space. For instance, mammal species are nearer each other while being far away from artifact or food. Moreover, features of the same category cluster and form a distribution while neighboring categories have overlapping features and cannot be fully separated. As such, we propose to quantify category IRs based on CKA [18], which is a normalized Hilbert-Schmidt Independence Criterion (HSIC) that models statistical relations of two sets of features by mapping them into a Reproducing Kernel Hilbert Space (RKHS) [21].

Given a set of $b$ training examples of category $\mathcal{C}_i$, we compute a matrix $\mathbf{X}_i \in \mathbb{R}^{u \times b}$ where the $k$-th column indicates the feature of example $k$ that is output from the DNN's penultimate layer. Then we compute a kernel matrix $\mathbf{K}_i \in \mathbb{R}^{b \times b}$ with some positive definite kernel, e.g., a linear kernel for which $\mathbf{K}_i = \mathbf{X}_i^T \mathbf{X}_i$ where $T$ indicates matrix transpose. Besides the linear kernel, we can choose other kernels such as polynomial kernel and RBF kernel (*cf. Section A.1 for details*). The IR between $\mathcal{C}_i$ and $\mathcal{C}_j$ is defined as:

$$\text{IR}(\mathcal{C}_i, \mathcal{C}_j) = \frac{\text{HSIC}(\mathcal{C}_i, \mathcal{C}_j)}{\sqrt{\text{HSIC}(\mathcal{C}_i, \mathcal{C}_i)}\sqrt{\text{HSIC}(\mathcal{C}_j, \mathcal{C}_j)}}, \ \text{HSIC}(\mathcal{C}_i, \mathcal{C}_j) = \frac{1}{(b-1)^2}\text{tr}(\mathbf{K}_i \mathbf{H} \mathbf{K}_j \mathbf{H}). \quad (1)$$

Here $\mathbf{H} = \mathbf{I} - \frac{1}{b}\mathbf{1}\mathbf{1}^T$ is the centering matrix where $\mathbf{I}$ indicates the identity matrix and $\mathbf{1}$ indicates an all-one vector; $\text{tr}$ indicates matrix trace. $\text{IR}(\mathcal{C}_i, \mathcal{C}_j) \in [0, 1]$ is invariant to isotropic scaling and orthogonal transformation. Note that the cost to compute the IRs can be neglected since we only need to compute them once beforehand. As the teacher is more knowledgeable, we compute category interrelations using the teacher model, which is indicated by $\text{IR}^{\mathcal{T}}(\mathcal{C}_i, \mathcal{C}_j)$.

Besides CKA, cosine similarity between the prototypes of two categories can also be used to quantify IRs. In practice, the prototype of one category can be computed as the average of the features of the category's examples. Alternatively, the weight vectors associated with the softmax classifier of a DNN model can be regarded as prototypes of individual categories [22].

**Loss function.**  Given an input image (instance), we let $\mathbf{z} = [z_i] \in \mathbb{R}^n$ be the corresponding logits of a DNN model where $i \in S_n \triangleq \{1, \cdots, n\}$ indicates the index of $i$-th category. The predicted category probability $\mathbf{p} = [p_i]$ is computed via the softmax function $\sigma$ with a temperature $\tau$, i.e., $p_i = \sigma\left(\frac{\mathbf{z}}{\tau}\right)_i \triangleq \exp(z_i/\tau)/\sum_{j \in S_n} \exp(z_j/\tau)$. We denote by $\mathbf{p}^{\mathcal{T}}$ and $\mathbf{p}^{\mathcal{S}}$ the predicted category

probabilities of the teacher and student models, respectively. The classical KD [2] is an instance-wise method, which measures the discrepancy between $\mathbf{p}^{\mathcal{T}}$ and $\mathbf{p}^{\mathcal{S}}$ given the same input image:

$$\mathrm{D_{KL}}(\mathbf{p}^{\mathcal{T}}\|\mathbf{p}^{\mathcal{S}}) = \sum_i p_i^{\mathcal{T}} \log \left( p_i^{\mathcal{T}}/p_i^{\mathcal{S}} \right). \qquad (2)$$

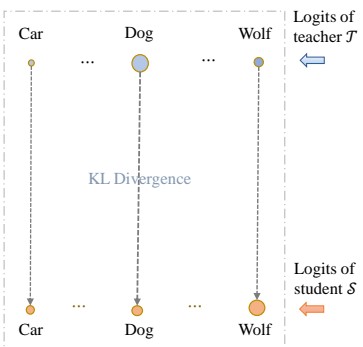

KL-Div (2) only compares predicted probabilities corresponding to the same category between the teacher and student, essentially short of a mechanism to perform cross-category comparison, as shown in Figure 2. Though during gradient back-propagation probability of one category affects probabilities of other categories due to the softmax function, *this implicit effect is insignificant and, above all, cannot explicitly exploit rich knowledge of pairwise interrelations* as described in (1).

In contrast to KL-Div, WD performs cross-category comparison and thus naturally makes use of category interrelations, as shown in Figure 1b (left). We formulate discrete WD as an entropy regularized linear programming [23]:

Figure 2: KL-Div cannot perform cross-category comparison. Compare to WD in Figure 1b (left).

$$\mathrm{D_{WD}}(\mathbf{p}^{\mathcal{T}}, \mathbf{p}^{\mathcal{S}}) = \min_{q_{ij}} \sum_{i,j} c_{ij}q_{ij} + \eta q_{ij} \log q_{ij} \qquad (3)$$

$$\text{s.t.} \ \ q_{ij} \geq 0, \ \sum_j q_{ij} = p_i^{\mathcal{T}}, \ \sum_i q_{ij} = p_j^{\mathcal{S}}, \ i,j \in S_n,$$

where $c_{ij}$ and $q_{ij}$ respectively indicate transport cost per mass and the transport amount while moving probability mass from $p_i^{\mathcal{T}}$ to $p_j^{\mathcal{S}}$; $\eta$ is a regularization parameter. We define the cost $c_{ij}$ by converting the similarity measure IRs to a distance measure according to the commonly used Gaussian kernel [10, Chap. 6], i.e., $c_{ij} = 1 - \exp(-\kappa(1 - \mathrm{IR}^{\mathcal{T}}(\mathcal{C}_i, \mathcal{C}_j)))$, where $\kappa$ is a parameter that can control the degree of sharpening of IR. *The smaller* $\mathrm{IR}^{\mathcal{T}}(\mathcal{C}_i, \mathcal{C}_j)$ *in the feature space, the less cost is needed for transport between the two categories.* As such, the loss function of WKD-L is

$$\tilde{\mathcal{L}}_{\mathrm{WKD\text{-}L}} = \mathrm{D_{WD}}(\mathbf{p}^{\mathcal{T}}, \mathbf{p}^{\mathcal{S}}). \qquad (4)$$

Recent work [3] discloses target probability (i.e., the probability of the target category) and the non-target ones play different roles: the former concerns the difficulty of training examples while the latter containing prominent "dark knowledge". It has been shown that this separation is helpful to balance their roles and improves greatly over the classical KD [3; 4]. Inspired by them, we also consider a similar separation strategy. Let $t$ be index of target category and $\mathbf{z}_{\backslash t}^{\mathcal{T}} = \left[ z_i^{\mathcal{T}} \right] \in \mathbb{R}^{n-1}, i \in S_n \backslash \{t\}$ be the teacher's logits of non-target categories. We normalize $\mathbf{z}_{\backslash t}^{\mathcal{T}}$ as previously, obtaining the teacher's non-target probabilities $\mathbf{p}_{\backslash t}^{\mathcal{T}} = \left[ p_i^{\mathcal{T}} \right]$. In this case, our loss functions consist of two terms:

$$\mathcal{L}_{\mathrm{WKD\text{-}L}} = \lambda \mathrm{D_{WD}}(\mathbf{p}_{\backslash t}^{\mathcal{T}}, \mathbf{p}_{\backslash t}^{\mathcal{S}}) + \mathcal{L}_\mathrm{t}, \ \ \mathcal{L}_\mathrm{t} = -\sigma(\mathbf{z}^{\mathcal{T}})_t \log \sigma(\mathbf{z}^{\mathcal{S}})_t, \qquad (5)$$

where $\lambda$ is the weight.

## 2.2 Continuous WD for Feature Distillation

As the features output from intermediate layers of DNN are of high dimension and small size, the non-parametric methods, e.g., histogram and kernel density estimation, are infeasible. Therefore, we use one of the widely used parametric methods (i.e., Gaussian) for distribution modeling.

**Feature distribution modeling.** Given an input image, let us consider feature maps output by some intermediate layer of a DNN model, whose spatial height, width and channel number are $h$, $w$ and $l$, respectively. We reshape the feature maps to a matrix $\mathbf{F} \in \mathbb{R}^{l \times m}$ where $m = h \times w$ and the $i$-th column $\mathbf{f}_i \in \mathbb{R}^l$ indicates a spatial feature. For these features, we estimate the 1st-moment $\boldsymbol{\mu} = \frac{1}{m} \sum_i \mathbf{f}_i$ and the 2nd-moment $\boldsymbol{\Sigma} = \frac{1}{m} \sum_i (\mathbf{f}_i - \boldsymbol{\mu})(\mathbf{f}_i - \boldsymbol{\mu})^T$. We model feature distribution of the input image by a Gaussian with mean vector $\boldsymbol{\mu}$ and covariance matrix $\boldsymbol{\Sigma}$ as its parameters:

$$\mathcal{N}(\boldsymbol{\mu}, \boldsymbol{\Sigma}) = \frac{1}{|2\pi\boldsymbol{\Sigma}|^{1/2}} \exp\left( -\frac{1}{2}(\mathbf{f} - \boldsymbol{\mu})^T \boldsymbol{\Sigma}^{-1} (\mathbf{f} - \boldsymbol{\mu}) \right), \qquad (6)$$

where $|\cdot|$ indicates matrix determinant.

We estimate Gaussian distribution of the teacher directly from its backbone network. For the student, as in previous arts [24; 25; 26], a projector is used to transform the features so that they are compatible in size with the features of the teacher. Then the transformed features produced by the projector are used to compute the student's distribution. We select Gaussian for distribution modeling as it is of maximal entropy for given the 1st- and 2nd-moments [19, Chap. 1] and has a closed form WD that is a Riemannian metric [20].

**Loss Function.** Let Gaussian $\mathcal{N}^{\mathcal{T}} \triangleq \mathcal{N}(\boldsymbol{\mu}^{\mathcal{T}}, \boldsymbol{\Sigma}^{\mathcal{T}})$ be feature distribution of the teacher. Similarly, we denote by $\mathcal{N}^{\mathcal{S}}$ the student's distribution. The continuous WD between the two Gaussians is defined as

$$\mathrm{D}_{\mathrm{WD}}(\mathcal{N}^{\mathcal{T}}, \mathcal{N}^{\mathcal{S}}) = \inf_q \int_{\mathbb{R}^l} \int_{\mathbb{R}^l} \|\mathbf{f}^{\mathcal{T}} - \mathbf{f}^{\mathcal{S}}\|^2 q(\mathbf{f}^{\mathcal{T}}, \mathbf{f}^{\mathcal{S}}) d\mathbf{f}^{\mathcal{T}} d\mathbf{f}^{\mathcal{S}}, \tag{7}$$

where $\mathbf{f}^{\mathcal{T}}$ and $\mathbf{f}^{\mathcal{S}}$ are Gaussian variables and $\|\cdot\|$ indicates Euclidean distance; the joint distribution $q$ is constrained to have marginals $\mathcal{N}^{\mathcal{T}}$ and $\mathcal{N}^{\mathcal{S}}$. Minimization of Eq. (7) leads to the following closed form distance [14]:

$$\mathrm{D}_{\mathrm{WD}}(\mathcal{N}^{\mathcal{T}}, \mathcal{N}^{\mathcal{S}}) = \mathrm{D}_{\mathrm{mean}}(\boldsymbol{\mu}^{\mathcal{T}}, \boldsymbol{\mu}^{\mathcal{S}}) + \mathrm{D}_{\mathrm{cov}}(\boldsymbol{\Sigma}^{\mathcal{T}}, \boldsymbol{\Sigma}^{\mathcal{S}}). \tag{8}$$

Here $\mathrm{D}_{\mathrm{mean}}(\boldsymbol{\mu}^{\mathcal{T}}, \boldsymbol{\mu}^{\mathcal{S}}) = \|\boldsymbol{\mu}^{\mathcal{T}} - \boldsymbol{\mu}^{\mathcal{S}}\|^2$ and $\mathrm{D}_{\mathrm{cov}}(\boldsymbol{\Sigma}^{\mathcal{T}}, \boldsymbol{\Sigma}^{\mathcal{S}}) = \mathrm{tr}(\boldsymbol{\Sigma}^{\mathcal{T}} + \boldsymbol{\Sigma}^{\mathcal{S}} - 2((\boldsymbol{\Sigma}^{\mathcal{T}})^{\frac{1}{2}} \boldsymbol{\Sigma}^{\mathcal{S}} (\boldsymbol{\Sigma}^{\mathcal{T}})^{\frac{1}{2}})^{\frac{1}{2}})$ where superscript $\frac{1}{2}$ indicates matrix square root. As the covariance matrices estimated from high-dimensional features are often ill-conditioned [27], we add a small positive number (1e-5) to the diagonals. We also consider diagonal covariance matrices, for which we have $\mathrm{D}_{\mathrm{cov}}(\boldsymbol{\Sigma}^{\mathcal{T}}, \boldsymbol{\Sigma}^{\mathcal{S}}) = \|\boldsymbol{\delta}^{\mathcal{T}} - \boldsymbol{\delta}^{\mathcal{S}}\|^2$, where $\boldsymbol{\delta}^{\mathcal{T}}$ is a vector of standard variances formed by the square roots of the diagonals of $\boldsymbol{\Sigma}^{\mathcal{T}}$. We later compare Gaussian (Full) and Gaussian (Diag) which have full and diagonal covariance matrices, respectively. To balance role of the mean and covariance, we introduce a mean-cov ratio $\gamma$ and define the loss as

$$\mathcal{L}_{\mathrm{WKD\text{-}F}} = \gamma \mathrm{D}_{\mathrm{mean}}(\boldsymbol{\mu}^{\mathcal{T}}, \boldsymbol{\mu}^{\mathcal{S}}) + \mathrm{D}_{\mathrm{cov}}(\boldsymbol{\Sigma}^{\mathcal{T}}, \boldsymbol{\Sigma}^{\mathcal{S}}). \tag{9}$$

We can use the strategy of spatial pyramid pooling [28; 29; 6] to enhance representation ability. Specifically, we partition the feature maps into a $k \times k$ spatial grid, compute a Gaussian for each cell of the grid and then match per cell the Gaussians of the teacher and student.

KL-Div [30] and symmetric KL-Div (i.e., Jeffreys divergence) [31], both having closed form expressions for Gaussians [32], can be used for knowledge transfer. However, they are not metrics [12], unaware of the geometry of the space of Gaussians [13] that is a Riemannian space. Conversely, $\mathrm{D}_{\mathrm{WD}}$ is a Riemannian metric that measures the intrinsic distance [20]. Note that $\mathrm{G}^2\mathrm{DeNet}$ [33] proposes a metric between Gaussians that leverages the geometry based on Lie group, which can be used to define distillation loss. Besides Gaussian, one can also use Laplace and exponential distributions for modeling feature distributions. Finally, though histogram or kernel density estimation are infeasible, one can still model feature distribution with probability mass function (PMF) and accordingly use discrete WD to define the distillation loss. *Details on these methods can be found in Section A.2.*

## 3   Related Works

We summarize KD methods related to ours and show their connections and differences in Table 1.

**KL-Div based knowledge distillation.**   Zhao et al. [3] disclose the classical KD loss [2] is a coupled formulation that limits its performance, and thereby propose a decoupled formulation (DKD) that consists of a binary logit loss for the target category and a multi-class logit loss for all non-target categories. Yang et al. [4] propose a normalized KD (NKD) method, which decomposes the classical KD loss into a combination of the target loss (like the widely used cross-entropy loss) and the loss of normalized non-target predictions. WTTM [5] introduces Rényi entropy regularizer without temperature scaling for student. In spite of competitive performance, they cannot explicitly exploit relations among categories. By contrast, our Wasserstein distance (WD) based method can perform cross-category comparison and thus exploit rich category interrelations.

**WD based knowledge distillation.**  The existing KD methods founded on WD [15; 16] mainly concern cross-instance matching for feature distillation, as shown in Figure 3 (left). Chen et al. [15] propose a Wasserstein Contrastive Representation Distillation (WCoRD) framework which involves

| Method | Logit Distillation | | | Feature Distillation | | |
|---|---|---|---|---|---|---|
| | Distribution | Dis-similarity | Category Interrelation | Distribution | Dis-similarity | Riemannian Metric |
| KD [2] DKD [3] NKD [4] WTTM [5] | Discrete | KL divergence | ✗ | – | | |
| WCoRD [15] | Discrete | Mutual Information | ✗ | Discrete | Wasserstein Distance | – |
| EMD+IPOT [16] | | – | | Discrete | Wasserstein Distance | – |
| WKD (ours) | Discrete | Wasserstein Distance | ✓ | Continuous (Gaussian) | Wasserstein Distance | ✓ |
| NST [35] | | – | | Spatial 2nd-moment | Frobenius | ✗ |
| ICKD-C [6] | | – | | Channel 2nd-moment | Frobenius | ✗ |
| VID [40] | Discrete | Mutual Information | ✗ | Continuous | Mutual Information | – |

Table 1: Comparison with related works.

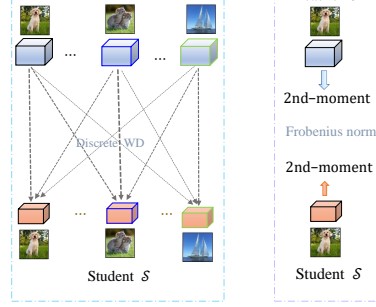

Figure 3: Diagrams of WCoRD /EMD+IPOT and NST/ICKD-C.

a global and a local contrastive loss. The former loss minimizes *mutual information* (via dual form of WD) between the distributions of the teacher and student; the latter loss minimizes *Wasserstein distance* for the penultimate layer only, where the set of features of the mini-batch images are matched between the teacher and student. Lohit et al. [16] independently propose a similar cross-instance matching method called EMD+IPOT, in which knowledge is transferred from all intermediate layers and discrete WD is computed via an inexact proximal optimal transport algorithm [34]. The differences of our work from them are twofold: (1) They fail to leverage category interrelations which our WKD-L can make full use of; (2) They are concerned with cross-instance matching based on discrete WD, while our WKD-F involves instance-wise matching with continuous WD.

**Other arts based on statistical modeling.** NST [35] is among the first to formalize feature distillation as a distribution matching problem, in which the student mimics the distributions of the teacher based on Maximum Mean Discrepancy (MMD). They show that the 2nd-order polynomial kernel performs best among the candidate kernels of MMD, and that the activation-based attention transfer (AT) [36] is a special case of NST. Yang et al. [37] propose a novel loss function, which transfers the statistics learned by the student back to the teacher based on adaptive instance normalization. Liu et al. [6] propose inter-channel correlations (ICKD-C) to model feature diversity and homology for better knowledge transfer. Both NST and ICKD-C can be regarded as Frobenius norm based distribution modeling via 2nd-moment along spatial- and channel-dimension, respectively, as shown in Figure 3 (right). However, they fail to utilize the geometric structure of the 2nd-moment matrices, which are symmetric positive definite (SPD) and form a Riemannian space [38; 39]. Ahn et al. [40] introduce variational information distillation (VID) based on mutual information. VID assumes that feature distributions are Gaussians, and its loss boils down to mean square loss (i.e., FitNet [24]) if Gaussians are further assumed to have unit variance.

# 4 Experiments

We evaluate WKD for image classification on ImageNet [41] and CIFAR-100 [42]. Also, we evaluate the effectiveness of WKD on self-knowledge distillation (Self-KD). Further, we extend WKD to object detection and conduct experiment on MS-COCO [43]. We train and test models with PyTorch framework [44], using a PC with an Intel Core i9-13900K CPU and GeForce RTX 4090 GPUs.

## 4.1 Experiment Setting

**Image classification.** ImageNet [41] contains 1,000 categories with 1.28M images for training, 50K images for validation and 100K for testing. In accordance with [25], we train the models for 100 epochs using SGD optimizer with a batch size of 256, a momentum of 0.9 and a weight decay of 1e-4. The initial learning rate is 0.1, divided by 10 at the 30th, 60th and 90th epochs, respectively. We use random resized crop and random horizontal clip for data augmentation. For WKD-L, we use POT library [45] for solving discrete WD with $\eta = 0.05$ and 9 iterations. For WKD-F, the projector has a bottleneck structure, i.e., a $1 \times 1$ Convolution (Conv) and a $3 \times 3$ Conv both with 256 filters followed by a $1 \times 1$ Conv with BN and ReLU to match the size of teacher's feature maps.

| Dis-similarity | Target←\|→Non-target | Top-1 | Δ |
|---|---|---|---|
| KL-Div | w/o [2] | 71.03 | – |
| | w/ [3] | 71.70 | +0.67 |
| | w/ [4] | 71.96 | +0.93 |
| WD | w/o | 72.04 | +1.01 |
| | w/ | **72.49** | **+1.46** |

(a) WD versus KL-Div.

| Dis-similarity | IR Method | | Top-1 | Δ |
|---|---|---|---|---|
| KL-Div [2] | – | | 71.03 | – |
| WD | CKA | Linear kernel | **72.49** | **+1.46** |
| | | Polynomial kernel | 72.10 | +1.07 |
| | | RBF kernel | 72.30 | +1.27 |
| | Cosine | Classifier weight | 72.19 | +1.16 |
| | | Class centroid | 72.02 | +0.99 |

(b) Different interrelation (IR) modelings.

Table 2: Ablation analysis of WKD-L for image classification (Acc, %) on ImageNet.

CIFAR-100 [42] contains 60K images of $32 \times 32$ pixels from 100 categories with 50K for training and 10K for testing. Following the setting of OFA [46], we conduct experiments across Convolutional Neural Networks (CNNs) and vision transformers. *Additional experiments within CNN architectures are given in Table 11 of Section C.6.* The images are upsampled to the resolution of 224x224. All models are trained for 300 epochs with a batch size of 512 and a cosine annealing schedule. For CNN-based students, we use SGD optimizer with an initial learning rate of 2.5e-2 and a weight decay of 2e-3. For transformer-based students, we use AdamW optimizer with an initial learning rate of 2.5e-4 and a weight decay of 2e-3.

**Object detection.** MS-COCO [43] is a commonly used benchmark for object detection that contains 80 categories. Following the common practice, we use standard split of COCO 2017 with 118K images for training and 5K images for validation. As in ReviewKD [29], we employ the framework of Faster-RCNN [47] with Feature Pyramid Network (FPN) [48] on Detectron2 platform [49]. As in previous arts [50; 29; 51], we use the detection models officially trained and released as the teachers, while training the student models whose backbones are initialized with the weights pre-trained on ImageNet. The student networks are trained in 180K iterations with a batch size of 8; the initial learning rate is 0.01, decayed by a factor of 0.1 at 120K and 160K iterations.

## 4.2 Dissection of WD-based Knowledge Distillation

We analyze key components of WKD-L and WKD-F on ImageNet. We adopt ResNet34 as a teacher and ResNet18 as a student (i.e., setting (a)), whose Top-1 Accuracies are 73.31% and 69.75%, respectively. *See Section C.1 for analysis on hyper-parameters, e.g., temperature and weight.*

### 4.2.1 Ablation of WKD-L

**How WD performs against KL-Div?** We compare WD to KL-Div with (w/) and without (w/o) separation of target probability in Table 2a. For the case of without separation, WD (w/o) improves over KL-Div (w/o) by 1.0%; for the case of with separation, WD (w/) outperforms non-trivially KL-Div (w/) based DKD and NKD. The comparison above clearly shows that (1) WD performs better than KL-Div in both cases and (2) the separation also matters for WD. As such, WD with separation of target probability is used across the paper.

**How to model category interrelations (IRs)?** Table 2b compares two methods for IR modeling, i.e., CKA and cosine. For the former, we assess different kernels while for the latter we evaluate prototypes with classifier weights or class centroids. We note all WD-based methods perform much better than the baseline of KL-Div. Overall, IR with CKA performs better than IR with cosine, indicating it has better capability to represent similarity among categories. For IR with CKA, RBF kernel is better than polynomial kernel, while linear kernel is the best so is used throughout.

### 4.2.2 Ablation of WKD-F

**Full covariance matrix or diagonal one?** As shown in Table 3a (3rd and 4th rows), for Gaussian (Full), WD performs better than $G^2$DeNet [33], which suggests that the former metric is more appropriate for feature distillation. When using WD, Gaussian (Diag) (5th row) produces higher accuracy than Gaussian (Full). We conjecture the reason is that high dimensionality of features makes estimation of full covariance matrices not robust [52]; in contrast, for Gaussians (Diag), we only need to estimate 1D variances for univariate data of single dimension. Besides, Gaussian (Diag) is much more efficient than Gaussian (Full). So we use Gaussians (Diag) throughout the paper.

**(a) Feature distribution modeling.**

| | Distribution | Dis-similarity | Top-1 | Δ |
|---|---|---|---|---|
| FitNet [24] | – | Frobenius | 70.53 | – |
| Parametric | Gaussian (Full) | WD | 72.37 | +1.84 |
| | | G²DeNet [33] | 72.23 | +1.70 |
| | Gaussian (Diag) | WD | **72.50** | **+1.97** |
| | | KL-Div | 71.75 | +1.22 |
| | | Sym KL-Div | 71.93 | +1.40 |
| | Laplace | KL-Div | 71.38 | +0.85 |
| | Exponential | KL-Div | 70.14 | -0.39 |
| | Spatial 1st-moment [35] | Euclidean | 70.56 | +0.03 |
| | Spatial 2nd-moment [35] | Frobenius | 71.14 | +0.61 |
| | Channel 1st-moment | Euclidean | 72.04 | +1.51 |
| | Channel 2nd-moment [6] | Frobenius | 71.59 | +1.06 |
| Non-parametric | PMF | WD | 71.57 | +1.04 |

**(b) Different matching strategies.**

| Method | WD | Matching Strategy | Top-1 | Δ |
|---|---|---|---|---|
| FitNet [24] | – | Instance-wise | 70.53 | – |
| WCoRD [15] | Discrete | Cross-instance | 71.49 | +0.96 |
| EMD+IPOT [16] | | | 70.46 | -0.07 |
| WKD-F | Continuous | Instance-wise | **72.50** | **+1.97** |

**(c) Distillation position and grid scheme.**

| | Stage | | Grid | | Top-1 | Δ |
|---|---|---|---|---|---|---|
| | Conv_4x | Conv_5x | 1×1 | 2×2 | | |
| FitNet [24] | | ✓ | ✓ | | 70.53 | – |
| WD | ✓ | | ✓ | | 71.52 | +0.99 |
| | | ✓ | ✓ | | **72.50** | **+1.97** |
| | | ✓ | | ✓ | 72.40 | +1.87 |
| | ✓ | ✓ | ✓ | | 72.44 | +1.91 |

Table 3: Ablation analysis of WKD-F for image classification (Acc, %) on ImageNet.

**How to model distributions?** In Table 3a, we compare different parametric methods for knowledge distillation, including Gaussian, Laplace, exponential distribution, as well as separate $1st$-moment and $2nd$-moment. Note that NST [35] adopts spatial 1st- and 2nd-moment while ICKD-C [6] uses channel $2nd$-moment. Besides, we compare to non-parametric method based on PMF.

*For Gaussians (Diag)*, KL-Div and symmetric (Sym) KL-Div produce similar accuracies which are both lower than WD. The reason may be that KL-related divergences are not intrinsic distances, failing to exploit geometric structure of the manifold of Gaussians. For *statistical moments*, we see that channel-wise moments perform better than spatial-wise ones. For channel-wise representations, 1st-moment outperforms 2nd-moment, suggesting that the mean plays a more important role. At last, the *non-parametric* method of PMF underperforms the parametric method of Gaussians.

Besides Gaussian (Diag), We can also use univariate *Laplace or exponential functions* to model distributions of each component of features. For them KL-Div can be computed in closed-form [53] but WD is an unsolved problem. When using KL-Div, Gaussian (Diag) achieves better performance than both Laplace and exponential distributions, highlighting Gaussian as a more suitable option among these parametric alternatives. Further, the Gaussian (Diag) combined with WD yields superior performance compared to KL-Div, suggesting advantage of the Riemannian metric.

**Instance-wise or cross-instance matching?** Our WKD-F is an instance-wise matching method based on continuous WD, while WCoRD and EMD+IPOT concern cross-instance matching for a mini-batch of images based on discrete WD. As seen in Table 3b, WCoRD produces an accuracy much higher than EMD+IPOT, which may be attributed to its extra global contrast loss based on mutual information; WKD-F outperforms the two counterparts by a large margin of 1.0%, which suggests the advantage of our strategy. Note that our WKD-F runs remarkably faster than the two counterparts that rely on optimization algorithm to solve discrete WD.

**Distillation position and grid scheme.** We evaluate in Table 3c the effect of position at which we perform distribution matching and that of different grid schemes. From the 3rd and 4th rows, we see that the last stage of Conv_5x performs much better than Conv_4x, indicating high-level features are more suitable for knowledge transfer. Comparing the 4th and 5th rows, we see $2\times2$ grid does not improve over $1\times1$ grid. Lastly, combination of features of Conv_4x and Conv_5x bring no further gains. Therefore, we use features of Conv_5x and $1\times1$ grid for classification on ImageNet.

## 4.3 Image Classification on ImageNet

Table 4 compares to existing works in two settings. Setting (a) involves homogeneous architecture, where the teacher and student networks are ResNet34 and ResNet18 [9], respectively; setting (b) concerns heterogeneous architecture, in which we set the teacher as ResNet50 and the student as MobileNetV1 [57]. *Refer to Section C.2 for hyper-parameters in Settings (a) and (b).*

For logit distillation, we compare our WKD-L with KD [2], DKD [3], NKD [4], CTKD [54] and WTTM [5]. Our WKD-L performs better than the classical KD and all its variants in both settings. Particularly, our WKD-L outperforms WTTM, a very strong variant of KD, which additionally introduces a sample-adaptive weighting method. This suggests Wasserstein distance that performs

| Setting | $\mathcal{T}$ | $\mathcal{S}$ | Logit | | | | | | Feature | | | | | Logit + Feature | | | | |
|---|---|---|---|---|---|---|---|---|---|---|---|---|---|---|---|---|---|---|
| | | | KD [2] | DKD [3] | NKD [4] | CTKD [54] | WTTM [5] | WKD-L (ours) | FitNet [24] | CRD [25] | Review -KD [29] | CAT [55] | WKD-F (ours) | CRD+ KD [25] | DPK [7] | FCFD [8] | KD-Zero[56] | WKD-L+ WKD-F (ours) |
| (a) Top-1 | 73.31 | 69.75 | 71.03 | 71.70 | 71.96 | 71.51 | 72.19 | **72.49** | 70.53 | 71.17 | 71.61 | 71.26 | **72.50** | 71.38 | 72.51 | 72.25 | 72.17 | **72.76** |
| (a) Top-5 | 91.42 | 89.07 | 90.05 | 90.41 | – | 90.47 | | **90.75** | 89.87 | 90.13 | 90.51 | 90.45 | **91.00** | 90.49 | 90.77 | 90.71 | 90.46 | **91.08** |
| (b) Top-1 | 76.16 | 68.87 | 70.50 | 72.05 | 72.58 | – | 73.09 | **73.17** | 70.26 | 71.37 | 72.56 | 72.24 | **73.12** | – | 73.26 | 73.26 | 73.02 | **73.69** |
| (b) Top-5 | 92.86 | 88.76 | 89.80 | 91.05 | – | – | – | **91.32** | 90.14 | 90.41 | 91.00 | 91.13 | **91.39** | – | 91.17 | 91.24 | 91.05 | **91.63** |

Table 4: Image classification results (Acc, %) on ImageNet. In setting (a), the teacher ($\mathcal{T}$) and student ($\mathcal{S}$) are ResNet34 and ResNet18, respectively, while setting (b) consists of a teacher of ResNet50 and a student of MobileNetV1. We refer to *Table 10 in Section C.4 for additional comparison to competitors with different setups.*

cross-category comparison is superior to category-to-category KL-Div. For feature distillation, we compare to FitNet [24], CRD [25], ReviewKD [29] and CAT [55]. Our WKD-F improves ReviewKD, previous top-performer, by $\sim$0.9% in the setting (a) and $\sim$0.6% in the setting (b) in terms of top-1 accuracy; this comparison indicates that, for knowledge transfer, matching of Gaussian distributions is better than matching of features. Finally, combination of WKD-L and WKD-F further improves and outperforms strong competitors, including CRD+KD [25], DPK [7], FCFD [8] and KD-Zero [56]. *More results of combination about WKD-L or WKD-F can be found in Table 9.*

Table 5 compares in Setting (a) latency of different methods with a batch size of 256 using a GeForce RTX 4090. For logit distillation, the latency of WKD-L is $\sim$1.3 times larger than KL-Div based methods, due to the optimization procedure to solve discrete WD. WKD-F has a latency on par with KL-Div based methods, while running $\sim$1.6 faster than ReviewKD and $\sim$1.2 faster than EMD+IPOT; this is because WKD-F only involves mean vectors and variance vectors, leading to negligibly additional cost. Finally, combination of WKD-L and WKD-F has larger latency but better performance than ICKD-C, and meanwhile is more efficient than state-of-the-art FCFD.

| Strategy | Method | Top-1 (%) | Params (M) | Latency (ms) |
|---|---|---|---|---|
| Logit | KD [2] | 71.03 | 0 | 215 |
| | NKD [4] | 71.96 | 0 | 214 |
| | WKD-L (Ours) | 72.49 | 0 | 280 |
| Feature | ReviewKD [29] | 71.61 | 7.2 | 349 |
| | EMD+IPOT [16] | 70.46 | 0.25 | 258 |
| | WKD-F (Ours) | 72.50 | 0.81 | 207 |
| Logit + Feature | FCFD [8] | 72.25 | 5.98 | 303 |
| | ICKD-C [6] | 72.19 | 0.33 | 222 |
| | WKD-L+ WKD-F (ours) | 72.76 | 0.81 | 292 |

Table 5: Training latency on ImageNet.

## 4.4 Image Classification on CIFAR-100

We evaluate WKD in the settings where the teacher is a CNN and the student is a Transformer or vice versa. We use CNN models including ResNet (RN) [9], MobileNetV2 (MNV2) [58] and ConvNeXt [59], as well as vision transformers that involve ViT [60], DeiT [61], and Swin Transformer [62]. *The setting of hyper-parameters can be found in Section C.5.*

For logit distillation, we compare WKD-L to KD [2], DKD [3], DIST [63] and OFA [46]. As shown in Table 6, WKD-L consistently outperforms state-of-the-art OFA for transferring knowledge from Transformers to CNNs or vice versa. For feature distillation, we compare to FitNet [24], CC [64], RKD [65] and CRD [25]. WKD-F ranks first across the board; notably, it significantly outperforms the previous best competitors by 2.1%–3.4% in four out of five settings. We attribute superiority of WKD-F to our distribution modeling and matching strategies, i.e., Gaussians and Wasserstein distance. We posit that, for knowledge transfer across CNNs and Transformers that yield very distinct features [46], WKD-F is more suitable than raw feature comparisons as in FitNet and CRD.

| Teacher (Acc) | Student (Acc) | Logit | | | | | Feature | | | | |
|---|---|---|---|---|---|---|---|---|---|---|---|
| | | KD [2] | DKD [3] | DIST [63] | OFA [46] | WKD-L (ours) | FitNet [24] | CC [64] | RKD [65] | CRD [25] | WKD-F (ours) |
| Transformer→CNN | | | | | | | | | | | |
| Swin-T (89.26) | RN18 (74.01) | 78.74 | 80.26 | 77.75 | 80.54 | **81.42**±0.22 | 78.87 | 74.19 | 74.11 | 77.63 | **81.57**±0.12 |
| ViT-S (92.04) | RN18 (74.01) | 77.26 | 78.10 | 76.49 | 80.15 | **80.81**±0.21 | 77.71 | 74.26 | 73.72 | 76.60 | **81.12**±0.24 |
| ViT-S (92.04) | MNV2 (73.68) | 72.77 | 69.80 | 72.54 | 78.45 | **79.04**±0.05 | 73.54 | 70.67 | 68.46 | 78.14 | **79.11**±0.07 |
| CNN→Transformer | | | | | | | | | | | |
| ConvNeXt-T (88.41) | DeiT-T (68.00) | 72.99 | 74.60 | 73.55 | 75.76 | **76.11**±0.18 | 60.78 | 68.01 | 69.79 | 65.94 | **73.27**±0.22 |
| ConvNeXt-T (88.41) | Swin-P (72.63) | 76.44 | 76.80 | 76.41 | 78.32 | **78.94**±0.17 | 24.06 | 72.63 | 71.73 | 67.09 | **74.80**±0.13 |

Table 6: Image classification results (Top-1 Acc, %) on CIFAR-100 across CNNs and Transformers.

### 4.5 Self-Knowledge Distillation on ImageNet

We implement our WKD in the framework of Born-Again Network (BAN) [66] for self-knowledge distillation (Self-KD). Specifically, we first train an initial model $S_0$ using ground truth labels. Then we distill, using WKD-L, the knowledge of $S_0$ into a student model $S_1$ with the same architecture as $S_0$. For the sake of simplicity, we do not perform multi-generation distillation, such as training a student model $S_2$ with $S_1$ as the teacher, etc.

We conduct experiments with ResNet18 on ImageNet, where the hyper-parameters are consistent with those in Setting (a). As shown in Table 7, BAN achieves competitive accuracy that is comparable to state-of-the-art results. Our method achieves the best result, outperforming BAN by $\sim 0.9\%$ in Top-1 accuracy and the second-best (i.e., USKD) by 0.6%. This comparison demonstrates that our WKD can well generalize to self-knowledge distillation.

| Method | Self-KD | Dis-similarity | Top-1 |
|---|---|---|---|
| Standard train | $\times$ | NA | 69.75 |
| Tf-KD [67] | ✓ | KL-Div | 70.14 |
| FRSKD [68] | ✓ | KL-Div | 70.17 |
| Zipf's KD [69] | ✓ | KL-Div | 70.30 |
| USKD [4] | ✓ | KL-Div | 70.75 |
| BAN [66] | ✓ | KL-Div | 70.50 |
| WKD-L | ✓ | WD | **71.35** |

Table 7: Self-KD results (Acc, %) on ImageNet with ResNet18.

### 4.6 Object Detection on MS-COCO

We extend WKD to object detection in the framework of Faster-RCNN [47]. For WKD-L, we use the classification branch in the detection head for logit distillation. For WKD-F, we transfer knowledge from features straightly fed to the classification branch, i.e., features output by the RoIAlign layer, and choose a $4\times4$ spatial grid for computing Gaussians. *Implementation details, ablation of key components, and extra experiments are given in Section E of Appendix.*

We compare with existing methods in two settings, as shown in Table 8. In RN101→RN18 setting, the teacher is ResNet101 and the student is ResNet18; in RN50→MNV2, the teacher and student are ResNet50 and MobileNetV2 [58], respectively. For logit distillation, our WKD-L significantly outperforms the classical KD [2] and is slightly better than DKD [3]. For feature distillation, we compare with FitNet, FGFI [50], ICD [51] and ReviewKD [29]; our WKD-F improves ReviewKD, the previous top feature distillation performer, by a non-trivial margin in both settings. Finally, by combining WKD-L and WKD-F, we achieve performance better than DKD+ReviewKD [3]. When additional bounding-box regression is used for knowledge transfer, our WKD-L+WKD-F improves further, outperforming previous state-of-the-art FCFD [8].

| Faster RCNN-FPN | | RN101→RN18 | | | RN50→MNV2 | | |
|---|---|---|---|---|---|---|---|
| | | mAP | $AP_{50}$ | $AP_{75}$ | mAP | $AP_{50}$ | $AP_{75}$ |
| Strategy | Teacher | 42.04 | 62.48 | 45.88 | 40.22 | 61.02 | 43.81 |
| | Student | 33.26 | 53.61 | 35.26 | 29.47 | 48.87 | 30.90 |
| Logit | KD [2] | 33.97 | 54.66 | 36.62 | 30.13 | 50.28 | 31.35 |
| | DKD [3] | 35.05 | 56.60 | 37.54 | 32.34 | 53.77 | 34.01 |
| | WKD-L (Ours) | **35.24** | **56.73** | **37.91** | **32.48** | **53.85** | **34.21** |
| Feature | FitNet [24] | 34.13 | 54.16 | 36.71 | 30.20 | 49.80 | 31.69 |
| | FGFI [50] | 35.44 | 55.51 | 38.17 | 31.16 | 50.68 | 32.92 |
| | ICD [51] | 35.90 | 56.02 | 38.75 | 32.88 | 52.56 | 34.93 |
| | ReviewKD [29] | 36.75 | 56.72 | 34.00 | 33.71 | 53.15 | 36.13 |
| | WKD-F (Ours) | **37.21** | **57.32** | **40.15** | **34.47** | **54.67** | **36.85** |
| Logit + Feature | DKD+ ReviewKD [3] | 37.01 | 57.53 | 39.85 | 34.35 | 54.89 | 36.61 |
| | WKD-L+ WKD-F (ours) | **37.49** | **57.76** | **40.39** | **34.80** | **55.27** | **37.28** |
| | FCFD$^\dagger$ [8] | 37.37 | 57.60 | 40.34 | 34.97 | 55.04 | 37.51 |
| | WKD-L+ WKD-F$^\dagger$ (ours) | **37.79** | **57.95** | **41.08** | **35.48** | **55.21** | **38.45** |

Table 8: Object detection results on MS-COCO. $^\dagger$Additional *bounding-box regression* is used.

## 5 Conclusion

The Wasserstein distance (WD) has shown evident advantages over KL-Div in several fields such as generative models [11]. However, in knowledge distillation, KL-Div is still dominant and it is unclear whether WD will outperform. We argue that earlier attempts on knowledge distillation based on WD fail to unleash the potential of this metric. Hence, we propose a novel methodology of WD-based knowledge distillation, which can transfer knowledge from both logits and features. Extensive experiments have demonstrated that discrete WD is a very promising alternative of predominant KL-Div in logit distillation, and that continuous WD can achieve compelling performance for transferring knowledge from intermediate layers. Nevertheless, our methods have limitations. Specifically, WKD-L is more expensive than KL-Div based logit distillation methods, while WKD-F assumes features follow Gaussian distribution. *We refer to Section F in Appendix for detailed discussion on limitations and future research.* Finally, we hope our work can shed light on the promise of WD and inspire further interest in this metric in knowledge distillation.

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

# A    Implementation Details on WKD

## A.1    Interrelations (IRs) among Category for WKD-L

**Visualization of category interrelations.**    We select in random fifty images from each of one hundred categories randomly chosen on the training set of ImageNet. Then we feed them to a pre-trained ResNet50 model and extract from the penultimate layer features that are projected to 2D space using t-SNE. The 2D embedding points are shown in Figure 4a where different categories are indicated by different colors. For intuitive understanding, inspired by karpathy[1], we display features by the corresponding images at their nearest 2D embedding locations in Figure 4b. It can be seen that the categories exhibit complex topological relations (distances) in the feature space, e.g., mammal species are nearer each other while far from artifact or food. The relations encode abundant information and are beneficial for knowledge distillation. In addition, the features of the same category cluster and form a (unknown) distribution that often overlaps with those of neighboring categories. The observation suggests that it is more desirable to model the interrelations with statistical method.

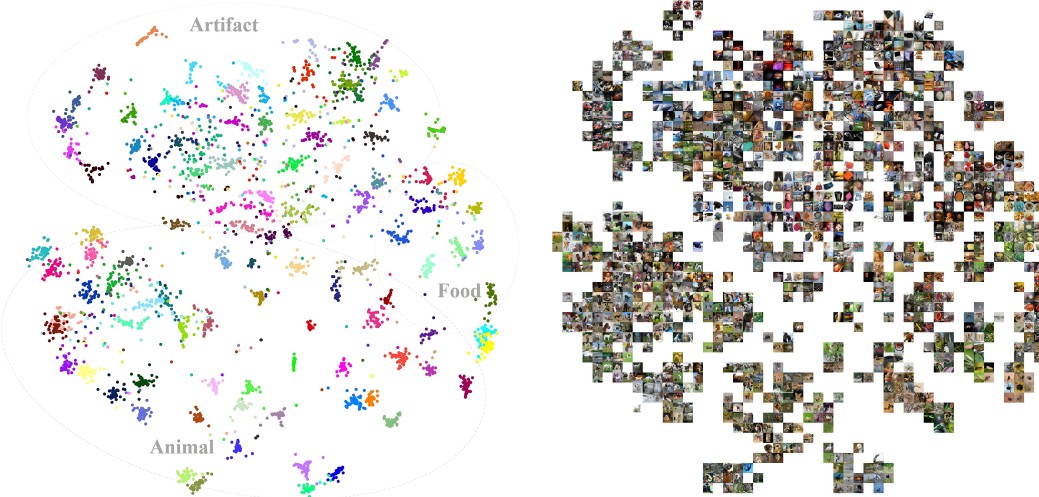

(a) Features are projected to 2D space using tSNE. Different categories are indicated by different colors.

(b) Features are displayed by the corresponding images at their nearest 2D embedding locations.

Figure 4: Visualization of interrelations among 100 categories in feature space. The categories exhibit complex topological relations, where features of the same category cluster and form a distribution that often overlaps with those of neighboring categories.

**Quantization of IRs with CKA.**    We propose to use CKA for modeling category IRs as it can effectively characterize similarity of deep representations [17]. CKA is normalized HSIC [18] that measures statistical dependence of random variables (features) by mapping them into a RKHS with some positive definite kernels. Recall that, for category $\mathcal{C}_i$, we have a matrix $\mathbf{X}_i \in \mathbb{R}^{u \times b}$ of $u$-dimensional features of $b$ training images. As such, for the commonly used kernels, we have linear kernel $\mathbf{K}_i^{\mathrm{lin}} = \mathbf{X}_i^T \mathbf{X}_i$, the polynomial kernel $\mathbf{K}_i^{\mathrm{poly}} = (\mathbf{X}_i^T \mathbf{X}_i + 1)^k$ where $k \in \{2, 3, 4\}$, and RBF kernel $\mathbf{K}_i^{\mathrm{rbf}} = \exp(-\frac{\mathbf{D}_i}{2\alpha^2 \mathrm{Med}(\mathbf{D}_i)})$ where the bandwidth $\alpha \in \{0.2, 0.4, 0.6\}$ and $\mathbf{D}_i = 2\left(\mathrm{diag}(\mathbf{X}_i^T \mathbf{X}_i)\mathbf{1}^T\right)_{\mathrm{sym}} - 2\mathbf{X}_i^T \mathbf{X}_i$. For a matrix $\mathbf{A}$, $\mathrm{Med}(\mathbf{A})$ denotes the median of all of its entries, $\mathrm{diag}(\mathbf{A})$ denotes a vector formed by its diagonals, and $(\mathbf{A})_{\mathrm{sym}} = \frac{1}{2}(\mathbf{A} + \mathbf{A}^T)$.

**Quantization of IRs with cosine similarity.**    Besides CKA, cosine similarity between the prototypes of two categories is used to quantify category interrelations. The prototype of a category can be naturally computed as *feature centroid of this category's training examples*, i.e., $\overline{\mathbf{x}}_i = \frac{1}{b}\mathbf{X}_i\mathbf{1}$. Alternatively, the weight vectors associated with the softmax classifier can be used as prototypes [22]. Specifically, if the weight matrix of the last FC layer is $\mathbf{W} \in \mathbb{R}^{u \times n}$ where $n$ is number of total classes, then its $i$-th column $\mathbf{w}_i$ can be regarded as the prototype of category $\mathcal{C}_i$, i.e., $\overline{\mathbf{x}}_i = \mathbf{w}_i$.

---

[1] https://cs.stanford.edu/people/karpathy/cnnembed

## A.2 Distributions Modeling for WKD-F

Recall that, for an input image, we have 3D feature maps output by some layer of a DNN, whose spatial height, width and channel number are $h$, $w$ and $l$, respectively. We reshape the feature maps to a matrix $\mathbf{F} \in \mathbb{R}^{l \times m}$ where $m = h \times w$; we denote the $i$-th column as a feature $\mathbf{f}_i \in \mathbb{R}^l$, while the $j$-th row (after transpose) as a feature $\widehat{\mathbf{f}}_j \in \mathbb{R}^m$. In WKD-F, we estimate channel 1st-moment $\boldsymbol{\mu} \in \mathbb{R}^l$ and 2nd-moment $\boldsymbol{\Sigma} \in \mathbb{R}^{l \times l}$:

$$\boldsymbol{\mu} = \frac{1}{m} \sum\nolimits_{i=1}^{m} \mathbf{f}_i, \ \boldsymbol{\Sigma} = \frac{1}{m} \sum\nolimits_{i=1}^{m} (\mathbf{f}_i - \boldsymbol{\mu})(\mathbf{f}_i - \boldsymbol{\mu})^T, \tag{10}$$

which are used to construct a parametric Gaussian $\mathcal{N}(\boldsymbol{\mu}, \boldsymbol{\Sigma})$. For measuring difference between Gaussians, we use Wasserstein distance (WD) that is a Riemannian metric. We prefer Gaussians (Diag) that have diagonal covariances to Gaussians (Full) that have full covariances, as they are much more efficient and have better performance as shown in Table 3a.

**$G^2$DeNet.** Different from WD, Wang et al. [33] propose a method called $G^2$DeNet, which considers Lie group of the space of Gaussians and embeds a Gaussian into the space of SPD matrices:

$$\mathcal{N}(\boldsymbol{\mu}, \boldsymbol{\Sigma}) \mapsto \begin{bmatrix} \boldsymbol{\Sigma} + \boldsymbol{\mu}\boldsymbol{\mu}^T & \boldsymbol{\mu} \\ \boldsymbol{\mu}^T & 1 \end{bmatrix}^{\frac{1}{2}}. \tag{11}$$

Thus the difference between two Gaussians is defined as the Euclidean distance of the Gaussian embeddings of the teacher and student models.

**ICKD-C and NST.** ICKD-C [6] uses *raw* channel 2nd-moment for exploring inter-channel correlations of features, i.e., $\mathbf{M} = \frac{1}{m} \sum_{i=1}^{m} \mathbf{f}_i \mathbf{f}_i^T$. Instead of channel-wise statistics, NST [35] estimates *raw* spatial 1st-moment $\widehat{\boldsymbol{\mu}} \in \mathbb{R}^m$ and *raw* spatial 2nd-moment $\widehat{\mathbf{M}} \in \mathbb{R}^{m \times m}$ for describing distributions:

$$\widehat{\boldsymbol{\mu}} = \frac{1}{l} \sum\nolimits_{j=1}^{l} \widehat{\mathbf{f}}_j, \ \widehat{\mathbf{M}} = \frac{1}{l} \sum\nolimits_{j=1}^{l} \widehat{\mathbf{f}}_j \widehat{\mathbf{f}}_j^T. \tag{12}$$

**KL-Div between Gaussians (Diag).** Let $\boldsymbol{\mu} = \begin{bmatrix} \mu_1, \cdots, \mu_l \end{bmatrix}^T$ be mean and $\boldsymbol{\delta} = \begin{bmatrix} \delta_1, \cdots, \delta_l \end{bmatrix}^T$ be variance of Gaussians (Diag). In this case, both KL-Div and symmetric KL-Div have closed forms:

$$\mathbf{D}_{\mathrm{KL}}(\mathcal{N}^{\mathcal{T}} \| \mathcal{N}^{\mathcal{S}}) = \frac{1}{2} \sum\nolimits_i \Big( \frac{\mu_i^{\mathcal{T}} - \mu_i^{\mathcal{S}}}{\delta_i^{\mathcal{S}}} \Big)^2 + \Big( \frac{\delta_i^{\mathcal{T}}}{\delta_i^{\mathcal{S}}} \Big)^2 - 2 \log \frac{\delta_i^{\mathcal{T}}}{\delta_i^{\mathcal{S}}} - 1, \tag{13}$$

$$\mathbf{D}_{\mathrm{Sym\ KL}}(\mathcal{N}^{\mathcal{T}} \| \mathcal{N}^{\mathcal{S}}) = \frac{1}{2} \sum\nolimits_i (\mu_i^{\mathcal{T}} - \mu_i^{\mathcal{S}})^2 \Big( \big( \frac{1}{\delta_i^{\mathcal{T}}} \big)^2 + \big( \frac{1}{\delta_i^{\mathcal{S}}} \big)^2 \Big) + \big( \frac{\delta_i^{\mathcal{S}}}{\delta_i^{\mathcal{T}}} \big)^2 + \big( \frac{\delta_i^{\mathcal{T}}}{\delta_i^{\mathcal{S}}} \big)^2 - 2. \tag{14}$$

Here $\mathcal{T}$ and $\mathcal{S}$ denote the teacher and student, respectively.

**KL-Div between Laplace distributions.** We assume components of feature $\mathbf{f} = [f_1, \cdots, f_l]^T \in \mathbb{R}^l$ are statistically independent. Then the Laplace distribution of $\mathbf{f}$ is $L(\boldsymbol{\mu}, \mathbf{v}) = \prod_i \frac{1}{2\nu_i} \exp(-\frac{|f_i - \mu_i|}{\nu_i})$, where $\boldsymbol{\mu}$ is the mean and $\mathbf{v} = [\nu_1, \cdots, \nu_l]^T$ is the scale parameter. The KL-Div takes the following form [53]:

$$\mathbf{D}_{\mathrm{KL}}(L^{\mathcal{T}} \| L^{\mathcal{S}}) = \sum\nolimits_i \log \frac{\nu_i^{\mathcal{S}}}{\nu_i^{\mathcal{T}}} + \frac{|\mu_i^{\mathcal{T}} - \mu_i^{\mathcal{S}}|}{\nu_i^{\mathcal{S}}} + \frac{\nu_i^{\mathcal{T}}}{\nu_i^{\mathcal{S}}} \exp\Big( -\frac{|\mu_i^{\mathcal{T}} - \mu_i^{\mathcal{S}}|}{\nu_i^{\mathcal{T}}} \Big) - 1. \tag{14}$$

**KL-Div between exponential distributions.** Under independence assumption, the exponential distribution of feature $\mathbf{f}$ is $E(\boldsymbol{\beta}) = \prod_i \beta_i \exp(-\beta_i f_i)$, where $\boldsymbol{\beta} = \begin{bmatrix} \beta_1, \cdots, \beta_l \end{bmatrix}^T$ is the rate parameter. For exponential distributions, the KL-Div can be written as [53]:

$$\mathbf{D}_{\mathrm{KL}}(E^{\mathcal{T}} \| E^{\mathcal{S}}) = \sum\nolimits_i \log \frac{\beta_i^{\mathcal{T}}}{\beta_i^{\mathcal{S}}} + \frac{\beta_i^{\mathcal{S}}}{\beta_i^{\mathcal{T}}} - 1. \tag{15}$$

**Non-parametric PMD.** Though non-parametric methods, e.g., histogram or kernel density estimation, are infeasible due to curse of dimensionality, we can still use probability mass function (PMF) for distribution modeling. Specifically, given a set of features $\mathbf{f}_i, i = 1, \ldots, m$, the PMF of the features is of the form:

$$\mathbf{p_f} = \sum\nolimits_{i=1}^{m} p_{\mathbf{f}_i} \psi(\mathbf{f}_i), \ p_{\mathbf{f}_i} = \frac{1}{m}, \tag{16}$$

where $\psi(\mathbf{f}_i)$ denotes Kronecker function that is equal to one if $\mathbf{f} = \mathbf{f}_i$ and zero otherwise. Let $\mathbf{p}_{\mathbf{f}}^{\mathcal{T}}/\mathbf{p}_{\mathbf{f}}^{\mathcal{S}}$ be the PMFs of the teacher/student. We can use discrete WD to measure their discrepancy, i.e., $D_{\mathrm{WD}}(\mathbf{p}_{\mathbf{f}}^{\mathcal{T}}, \mathbf{p}_{\mathbf{f}}^{\mathcal{S}})$; the transport cost $c_{ij}$ between two features $\mathbf{f}_i^{\mathcal{T}}$ and $\mathbf{f}_j^{\mathcal{S}}$ is computed as $1 - \cos(\mathbf{f}_i^{\mathcal{T}}, \mathbf{f}_j^{\mathcal{S}})$. Note that KL-Div is *inapplicable* to PMF as it cannot handle non-overlapping distributions [11].

It is worth noting that (1) the space of Gaussians is a Riemannian space, on which WD is an intrinsic distance [20] while KL-Div or its symmetric version are not [12] and thus are unaware of the geometry [13]; (2) the space of either channel or spatial 2nd-moments is a manifold of symmetric, positive definite matrices rather than a Euclidean space [38; 39], so the Frobenius norm is not an intrinsic distance [35; 6] and fails to exploit the geometric structure of the manifold.

## B Computational Complexity of WKD

The logit-based WKD-L is formulated as an entropy regularized linear programming, which can be solved fastly by Sinkhorn algorithm [23]. Let $n$ be the dimension of the predicted logits, the complexity of WKD-L can be written as $O(Dn^2 \log n)$ [70]. Here $D = \|\mathcal{C}\|_\infty^3 \epsilon$ is a constant, where $\|\mathcal{C}\|_\infty$ indicates the infinity norm of the transportation cost matrix $\mathcal{C} = [c_{ij}]$, and $\epsilon > 0$ indicates a prescribed error. In contrast, the computational complexity of KL-Div is $O(n)$. Despite its high complexity, WKD-L can be computed efficiently as Sinkhorn algorithm is highly suitable for parallel computation on GPU [23]. For feature-based WKD-F, the dominant cost lies in computation of means and variances. Given a set of $m$ features $\mathbf{f}_i$ of $l$-dimension, the means can be computed by global average pooling that takes $O(ml)$ time; the complexity of variances is also $O(ml)$, as it can be obtained by element-wise square operations followed by global average pooling.

## C Extra Experiment on Image Classification

### C.1 More Ablation Analysis of WKD

We adopt the setting (a) for ablation on ImageNet, in which the teacher is ResNet34 and the student is ResNet18.

**Hyper-parameters of WKD-L.** The total loss function of WKD-L consists of the cross-entropy loss $\mathcal{L}_{\mathrm{CE}}$, the target loss $\mathcal{L}_{\mathrm{t}}$ and WD-based logit loss $\mathcal{L}_{\mathrm{WKD\text{-}L}}$. Following [3; 4], we set the weights of the two former losses to 1 across the paper. As such, our hyper-parameters includes the temperature $\tau$, the weight $\lambda$ of $\mathcal{L}_{\mathrm{WKD\text{-}L}}$, the sharpening parameter $\kappa$ that controls smoothness of IRs for transport cost in WD, and the regularization parameter $\eta$ (set to 0.05) in discrete WD. As simultaneous optimization of them is computationally infeasible, we first analyze the effect of temperature by

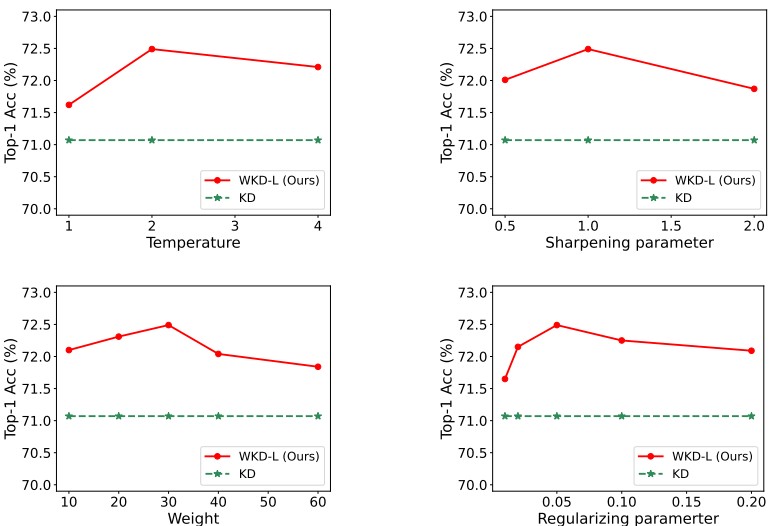

Figure 5: Analysis of hyper-parameters of WKD-L on ImageNet.

fixing the sharpening parameter and weight to 1 and 30, respectively. As seen in *Figure 5 (top left)*, across all temperatures WKD-L performs better than the baseline of KD [2] by large margins while achieving the highest accuracy when the temperature is 2. Next, we fix the temperature and weight, analyzing the effect of sharpening parameter. From *Figure 5 (top right)* we see that performance variation is not large against the sharpening parameter and the best performance is obtained when it is equal to 1. Subsequently, by fixing the best temperature and sharpening parameters, we evaluate the weight of WKD-L. As *Figure 5 (bottom left)* shows, its accuracy varies smoothly against the weight and achieves the best accuracy when the weight is 30. Finally, *Figure 5 (bottom right)* illustrates that the performance varies smoothly as a function of the regularization parameter $\eta$ over a reasonably wide range; for consistency, we set $\eta$ to 0.05 in all experiments throughout this paper.

**Hyper-parameters of WKD-F.** The loss function of WKD-F method contains the cross-entropy loss $\mathcal{L}_{\text{CE}}$ and WD-based feature loss $\mathcal{L}_{\text{WKD-F}}$. As in previous work, we set the weight of the cross-entropy loss to 1. Therefore, there are two hyper-parameters, i.e., the mean-cov ratio $\gamma$ and the weight of $\mathcal{L}_{\text{WKD-F}}$. We first analyze the effect of the former by fixing the latter to 2e-2. *Table 6 (left)* shows the performance as a function of the mean-cov ratio. We see that in the whole range of the mean-cov ratio WKD-F is clearly better than the baseline of FitNet [24]; the best accuracy is obtained when its value is 2, which suggests that the means play a more important role than the covariances. Next, we fix the mean-cov ratio and evaluate the effect of the weight. As seen in *Table 6 (right)*, WKD-F is much better than the baseline of FitNet across all weights and achieves the best performance when the weight is 0.02.

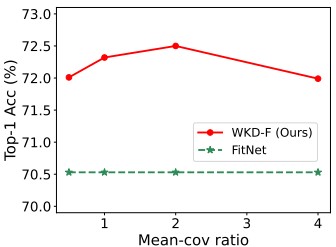 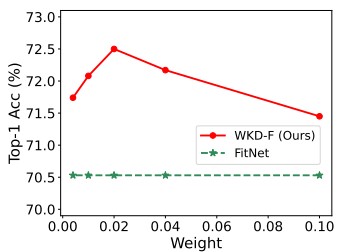

Figure 6: Analysis of hyper-parameters on WKD-F on ImageNet.

## C.2 Summary of Hyper-parameters on ImageNet

*The setting (a) involves homogeneous architecture*, where the teacher and student networks are ResNet34 and ResNet18, respectively. For WKD-L, the weights of $\mathcal{L}_{\text{CE}}$, $\mathcal{L}_{\text{t}}$ and $\mathcal{L}_{\text{WKD-L}}$ are 1, 1 and 30, respectively; the temperature $\tau = 2$ and sharping parameter $\kappa = 1$. For WKD-F, we set the weights of $\mathcal{L}_{\text{CE}}$ and $\mathcal{L}_{\text{WKD-F}}$ to 1 and 2e-2, respectively. We adopt features from Conv5_x and Gaussian (Diag) with mean-cov ratio $\gamma = 2$. *The setting (b) concerns heterogeneous architecture*, in which the teacher is ResNet50 and the student is MobileNetV1. In this setting, the weight of $\mathcal{L}_{\text{WKD-L}}$ is set to 25 while that of $\mathcal{L}_{\text{WKD-F}}$ is 1e-3, and the remaining hyper-parameters are identical to those in the setting (a).

## C.3 More Experiments on Combination of WKD-L or WKD-F

**How WKD complements other KD methods?** We combine WKD-F with a state-of-the-art logit-based knowledge distillation (i.e., NKD), and combine WKD-L with a state-of-the-art feature-based method (i.e., ReviewKD). The results are shown in Table 9a and Table 9b, respectively. We can see that NKD+WKD-F improves over individual NKD and WKD-F, which suggests our WKD-F is complementary to NKD. Notably, NKD+WKD-F is slightly inferior to WKD-L+WKD-F. In addition, WKD-L+ReviewKD improves over ReviewKD but underperforms WKD-L. We conjecture that, due to the large gap ($\sim 0.9\%$) between ReviewKD and WKD-L, the combination hurts WKD-L.

**Will separation of target probability in WKD-L still help when combined with WKD-F?** To answer this question, we integrate WKD-L without separation into WKD-F, which is called WKD-L (w/o)+WKD-F. From Table 9c, we can see that it is slightly inferior to WKD-L (w/)+WKD-F in which the separation scheme is used in WKD-L. This suggests that the benefit due to the separation scheme decreases but still persists in the logit+feature approach.

| Method | Top-1 | Top-5 |
|---|---|---|
| NKD [4] | 71.96 | – |
| WKD-F | 72.50 | 91.00 |
| NKD+WKD-F | **72.68** | **91.05** |
| WKD-L | 72.49 | 90.75 |
| WKD-L+WKD-F | **72.76** | **91.08** |

(a) Combination of NKD and WKD-F

| Method | Top-1 | Top-5 |
|---|---|---|
| WKD-L | 72.49 | 90.75 |
| ReviewKD [29] | 71.61 | 90.51 |
| WKD-L+ReviewKD | 72.32 | 90.63 |
| WKD-L | 72.49 | 90.75 |
| WKD-L+WKD-F | **72.76** | **91.08** |

(b) Combination of WKD-L and ReviewKD

| Method | Target←\|→Non-target | Top-1 | Top-5 |
|---|---|---|---|
| WKD-L | w/o | 72.04 | 90.42 |
| WKD-L | w/ | 72.49 | 90.75 |
| WKD-F | – | 72.50 | 91.00 |
| WKD-L+WKD-F | w/o | 72.70 | 91.07 |
| WKD-L+WKD-F | w/ | **72.76** | **91.08** |

(c) Effect of separation scheme on WKD-L+WKD-F.

Table 9: Analysis on different combinations of logit and feature distillation methods.

## C.4 Comparison to Competitors with Different Setups

In Table 4 of the main paper, we conduct comparison in the ordinary setting, which is formalized in CRD [25] and adopted by most methods. However, there exist some works which adopt Settings (a) and (b) but with non-trivially different setups, e.g., DIST [63], NKD [4], MGD [71], and DiffKD [72]. Specifically, *their students and/or teachers have performance higher than those in the ordinary settings*, which make them somewhat advantageous when comparing with the metric of Top-1 accuracy. For a fair comparison, we propose to use *gains of the distilled student over the vanilla student* in Top-1 accuracy, which is denoted by ▲. They can more faithfully indicate how much knowledge the student has learned from the teacher. The comparison results are shown in Table 10.

For *logit distillation*, WKD-L outperforms NKD, a strong KL-Div based variant, by 0.68% and 0.93% in setting (a) and setting (b), respectively. Instead of KL-Div, DIST matches predicted probabilities based on Pearson correlation coefficients, which exploits both instance-wise and cross-instance knowledge, in contrast to WKD-L that uses instance-wise knowledge only. Nevertheless, WKD-L performs better than it by 0.43% in setting (a) and a large margin of 1.19% in setting (b). For *feature distillation*, we compare to MGD that randomly masks the student's features and then forces the student to generate the teacher's features. The accuracies of WKD-F are over 1.0% higher than those of MGD in both settings. Regarding *logit+feature distillation*, we contrast with MGD+WSLD and DiffKD. DiffKD denoises the student features through a diffusion model trained by teacher features whose computation cost is high. WKD-L+WKD-F surpasses DiffKD by 0.55% and 1.33% in setting (a) and setting (b), respectively.

| Strategy | Method | Setting (a): ResNet34 → ResNet18 | | | | Setting (b): ResNet50 → MobileNetV1 | | | |
|---|---|---|---|---|---|---|---|---|---|
| | | Teacher | Vanilla student | Distilled student | ▲ | Teacher | Vanilla student | Distilled student | ▲ |
| Logit | NKD [4] | **73.62** | **69.90** | 71.96 | +2.06 | **76.55** | **69.21** | 72.58 | +3.37 |
| | DIST [63] | 73.31 | 69.76 | 72.07 | +2.31 | 76.16 | **70.13** | 73.24 | +3.11 |
| | WKD-L (ours) | 73.31 | 69.75 | 72.49 | **+2.74** | 76.16 | 68.87 | 73.17 | **+4.30** |
| Feature | MGD [71] | **73.62** | **69.90** | 71.58 | +1.68 | **76.55** | **69.21** | 72.35 | +3.14 |
| | WKD-F (our) | 73.31 | 69.75 | 72.50 | **+2.75** | 76.16 | 68.87 | 73.12 | **+4.25** |
| Logit + Feature | MGD+WSLD [71] | **73.62** | **69.90** | 71.80 | +1.90 | **76.55** | **69.21** | 72.59 | +3.38 |
| | DiffKD [72] | 73.31 | 69.76 | 72.22 | +2.46 | 76.16 | **70.13** | 73.62 | +3.49 |
| | WKD-L+WKD-F (ours) | 73.31 | 69.75 | 72.76 | **+3.01** | 76.16 | 68.87 | 73.69 | **+4.82** |

Table 10: Image classification results (Top-1 Acc, %) on ImageNet *between WKD and the competitors with different setups.* **Red** numbers indicate *the teacher/student model has non-trivially higher performance than the commonly used ones* formalized in CRD [25]. ▲ represents the gains of the distilled student over the vanilla student.

Additionally, MLKD [73] makes use of a stronger image augmentation, i.e., RandAugment [74] (cf. their implementation), for improving performance. Vanilla KD [75] studies the great potential of vanilla KD in a very different setting, in which optimizer with much longer epochs, diverse and very strong augmentations along with more regularization methods are used. In contrast, we and many of the state-of-the-art methods follow the ordinary setting formalized in CRD [25].

### C.5 Summary of Hyper-parameters for *across CNNs and Transformers* on CIFAR-100

For a fair comparison, *we follow OFA [46] and separately tune the hyper-parameters for different settings*. For WKD-L, we set the temperature to 2 and the sharpening parameter to 1, while searching the optimal weight of $\mathcal{L}_{\text{WKD-L}}$ in $[50, 200]$ with a step of 50. For WKD-F, the projector is simply a linear layer. We set mean-cov ratio to 2, and perform grid search for the weight of $\mathcal{L}_{\text{WKD-F}}$ in $\{1, 2, 4, 8\} \times 1\text{e-}2$. The spatial grid for computing Gaussians (Diag) is set to $1 \times 1$. We report average accuracy and standard deviation after three runs in Table 6; the results of all competing methods are duplicated from OFA.

### C.6 Knowledge Distillation *within CNN Architectures* on CIFAR-100

**Experimental Setup.** We follow the setting of CRD [25], where the networks cover ResNet [9], Wide-ResNet (WRN) [76], VGG [77], MobileNetV2 [58], and ShuffleNetV1 (SNV1) [78]. All models are trained for 240 epochs via the SGD optimizer with a batch size of 64, a momentum of 0.9 and a weight decay of 0.0005. The initial learning rate is 0.01 for MobileNetV2 and ShuffleNetV1 and 0.05 for the remaining networks, divided by 10 at the 150th, 180th, and 210th epochs, respectively. As in CRD [25], the projector is a $1 \times 1$ Conv or $4 \times 4$ transposed Conv both with BN and ReLU.

For a fair comparison, *we follow the practice of DKD [3], CAT [55], ReviewKD [29], FCFD [8] and WTTM [5]*, i.e., tuning hyper-parameters separately for different architectures. For WKD-L, we perform grid search for the weight of $\mathcal{L}_{\text{WKD-L}}$ in $[50, 800]$ with a step of 50, the temperature $\tau$ in $\{4, 8\}$ and the sharpening parameter $\kappa$ in $\{0.5, 1\}$. For WKD-F, we perform grid search for the weight of $\mathcal{L}_{\text{WKD-F}}$ in $\{1, 2, \ldots, 50\} \times 1\text{e-}2$ and mean-cov ratio $\gamma$ in $\{2, 3, \ldots, 8\}$. The spatial grid for computing Gaussians (Diag) is searched in $\{1 \times 1, 2 \times 2, 4 \times 4\}$. We report the average accuracy and standard deviation of our method after three runs.

**Results.** The comparison results are shown in Table 11. For logit distillation, WKD-L outperforms the classical KD by large margins in all settings, and surpasses the competitors in 5 out of 6 settings across homogeneous and heterogeneous architectures. For feature distillation, WKD-F invariably achieves better results than EMD-based counterparts (i.e., WCoRD and EMD+IPOT) and 2nd-moment based counterpart (i.e., NST); meanwhile, WKD-F is also very competitive, compared to other state-of-the-art methods. For logit+feature distillation, WKD-L+WKD-F ranks first in 4 out of 6 settings across the board. Overall, our methods have comparable or lower standard deviation, as opposed to the competing methods, which suggests that our method is statistically robust.

| Strategy | | Homogeneous | | | Heterogeneous | | |
|---|---|---|---|---|---|---|---|
| | Teacher | WRN40-2 | RN32x4 | VGG13 | WRN40-2 | VGG13 | RN50 |
| | Student | WRN40-1 | RN8x4 | VGG8 | SNV1 | MNV2 | MNV2 |
| Strategy | Teacher | 75.61 | 79.42 | 74.64 | 75.61 | 74.64 | 79.34 |
| | Student | 71.98 | 72.50 | 70.36 | 70.50 | 64.60 | 64.60 |
| Logit | KD [2] | 73.54±0.20 | 73.33±0.25 | 72.98±0.19 | 74.83±0.17 | 67.37±0.32 | 67.35±0.32 |
| | DIST [63] | 74.73±0.24 | 76.31±0.19 | – | – | – | 68.66±0.23 |
| | DKD [3] | 74.81 | 76.32 | 74.68 | 76.70 | 69.71 | 70.35 |
| | NKD [4] | – | 76.35 | 74.86 | – | **70.22** | 70.67 |
| | WTTM [5] | 74.58 | 76.06 | 74.44 | 75.42 | 69.16 | 69.59 |
| | WKD-L (Ours) | **74.84**±0.32 | **76.53**±0.14 | **75.09**±0.13 | **76.72**±0.09 | 70.21±0.24 | **71.10**±0.16 |
| Feature | FitNet [24] | 72.24±0.24 | 73.50±0.28 | 71.02±0.31 | 73.73±0.32 | 64.14±0.50 | 63.16±0.47 |
| | VID [40] | 73.30±0.13 | 73.09±0.21 | 71.23±0.06 | 73.61±0.12 | 65.56±0.42 | 67.57±0.28 |
| | CRD [25] | 74.14±0.22 | 75.51±0.18 | 73.94±0.22 | 76.05±0.14 | 69.73±0.42 | 69.11±0.28 |
| | ReviewKD [29] | **75.09** | 75.63 | 74.84 | 77.14 | **70.37** | 69.89 |
| | CAT [55] | 74.82 | **76.91** | 74.65 | 77.35 | 69.13 | 71.36 |
| | NST [35] | 72.24±0.22 | 73.30±0.28 | 71.53±0.13 | 74.89±0.25 | 58.16±0.26 | 64.96±0.44 |
| | WCoRD [15] | 74.73 | 75.95 | 74.55 | 76.32 | 69.47 | 70.45 |
| | EMD+IPOT[16] | – | 74.19 | 72.80 | – | – | – |
| | WKD-F (Ours) | 75.02±0.06 | 76.77±0.26 | **75.02**±0.18 | **77.36**±0.19 | 70.34±0.15 | **71.87**±0.35 |
| Logit + Feature | DPK [7] | 75.27 | – | 74.96 | – | – | – |
| | FCFD [8] | **75.46** | 76.62 | 75.22 | **77.99** | 70.65 | 71.00 |
| | DiffKD [72] | 74.09±0.09 | 76.72±0.15 | – | – | – | 69.21±0.27 |
| | ICKD-C [6] | 74.63 | 75.48 | 73.88 | – | – | – |
| | WKD-L+ WKD-F (Ours) | 75.35±0.13 | **77.28**±0.24 | **75.25**±0.15 | 77.50±0.32 | **70.68**±0.10 | **71.71**±0.28 |

Table 11: Image classification results (Top-1 Acc, %) on CIFAR-100 *within CNN architectures*.

# D  Visualization

## D.1  Visualization of Teacher-Student Discrepancies

Following [25; 3], we visualize the difference of correlation matrices of student and teacher logits. Figure 7a shows visualization results in two settings on CIFAR-100 [42]. In setting of ResNet32x4→ResNet8x4, the teacher and student are ResNet32x4 and ResNet8x4, respectively; the setting VGG13→VGG8 consists of a teacher of VGG13 and a student of VGG8. On the whole, in both settings the colors of WKD-L are much lighter than those of KD [2], which indicates WKD-L produces correlations matrices more similar to the teacher than KD. As the differences of correlation matrices capture inter-class correlations [25], smaller differences of WKD-L suggest better-informed cross-category relations than KL divergence can be learned.

In addition, we visualize whether WKD-F can learn distributions that are more similar to the teacher. To that end, we use high-level features output from the last Conv layer of a network model for computing distributions, as they encode the most discriminative information for classification. For each validation image of the $i$-th category, we compute WD between feature distributions of the teacher and student. Hence, for $n$ categories each with $k$ validation images, we obtain a $n \times k$ matrix of distribution matching. Figure 7b shows visualization of two settings on CIFAR-100. We can see that overall WKD-F demonstrates smaller discrepancies with the teacher than FitNet [24], suggesting it can better mimic the teacher's distributions.

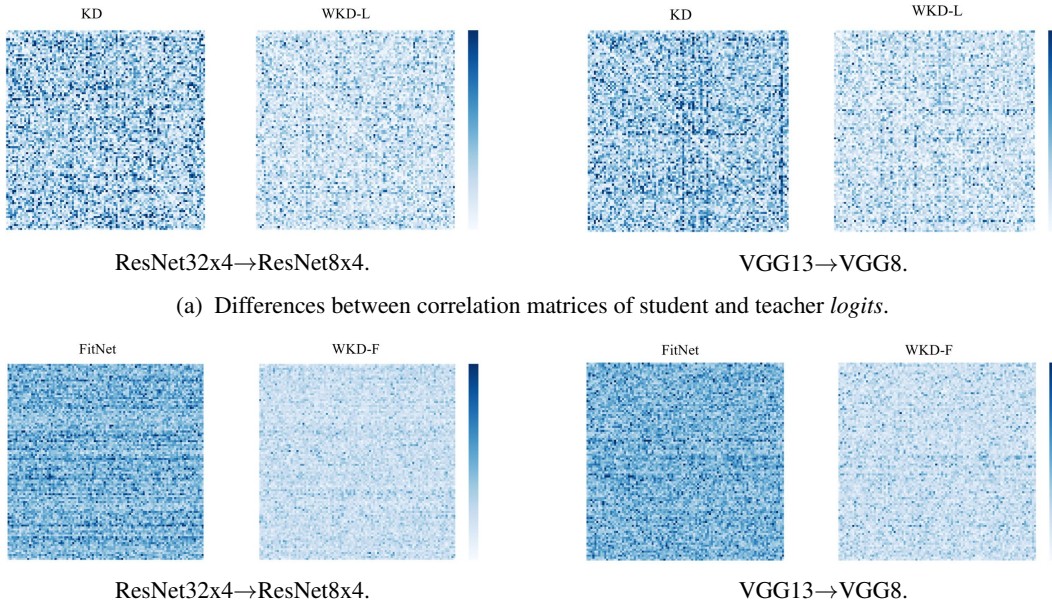

(a)  Differences between correlation matrices of student and teacher *logits*.

(b)  Differences between distributions of teacher and student *features*.

Figure 7:  Visualization of teacher-student discrepancies for WKD-L (a) and WKD-F (b). Darker color indicates larger difference.

## D.2  Visualization of Class Activation Maps (CAMs)

We use Grad-CAM [79] to compute class activation maps using features output from the last Conv layer. Figure 8 shows, for three example images, CAMs of different models, including the teacher, vanilla student, distilled models by KD, WKD-L, FitNet and WKD-F. It can be seen that WKD-L and WKD-F have more similar CAMs with the teacher than KD and FitNet, and localize more accurately the important regions of objects. The comparison suggests that WKD-L and WKD-F can learn features with better representation capability.

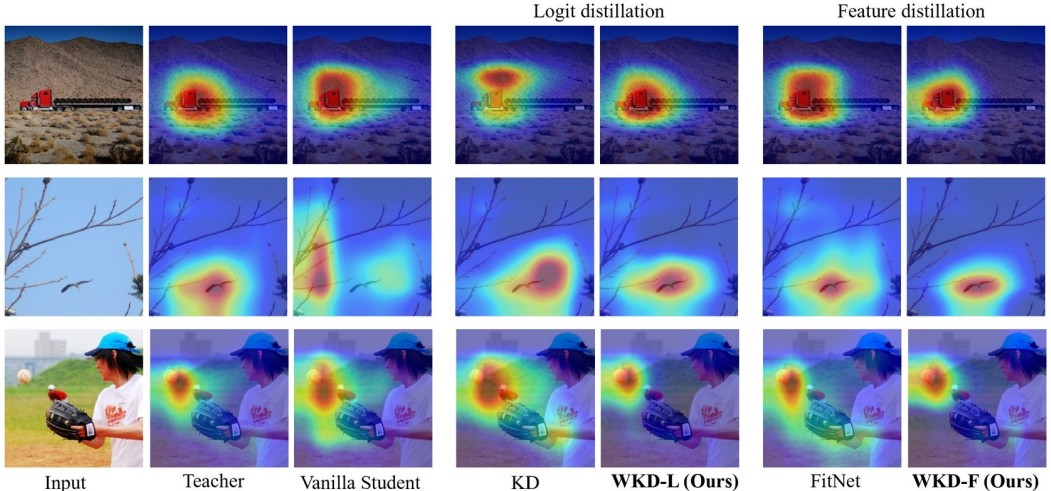

| Input | Teacher | Vanilla Student | KD | **WKD-L (Ours)** | FitNet | **WKD-F (Ours)** |

Figure 8: Visualization of different models via Grad-CAM.

# E Extra Experiment on Object Detection

The framework of Faster-RCNN [47] with Feature Pyramid Network (FPN) [48] is much more complicated than that of image classification. It contains the backbone network, Region Proposal Network (RPN), Feature Pyramid Network (FPN), and detection head consisting of a classification branch and a localization branch. For feature distillation, besides RoIAlign layer, FPN can also be used for knowledge transfer [29].

## E.1 Implementation details on COCO

For WKD-L, we use discrete WD to match the probabilities predicted by the classification branches of the teacher and student. We set the temperature $\tau = 1$ and sharpening parameter $\kappa = 2$, and set all the weights of $\mathcal{L}_{CE}$, $\mathcal{L}_{t}$ and $\mathcal{L}_{WKD-L}$ to 1 for RN101→RN18. All hyper-parameters of RN50→MV2 are the same as RN101→R18 but $\kappa$ that is set to 1. For WKD-F, we transfer knowledge from features straightly fed to the classification branch, i.e., features output by the RoIAlign layer. We let RoIAlign generate a high resolution of 18×18 feature maps to exploit more features, and choose a 4×4 spatial grid for computing Gaussians. For both settings, we set the weight of $\mathcal{L}_{CE}$ to 1, and that of $\mathcal{L}_{WKD-F}$ to 5e-3 with mean-cov ratio $\gamma = 2$; as in FCFD [8], we use a projector of 3×3 Conv with BN.

## E.2 Ablation Analysis of WKD-F for Object Detection

We analyze key component of WKD-F specific to object detection on MS-COCO [43] with ResNet101 as the teacher and ResNet18 as the student.

**Where to distill features: RoIAlign or FPN?** Among five stages (P2–P6) of FPN, we select a single P3 with a spatial grid of 16×16 for extracting Gaussians; this option produces the best result among the candidates, and combination of multiple stages brings us no benefit. We compare to the baseline of FitNet [24] and the results are shown in Table 12a. When distilling features of FPN, WKD-F significantly outperforms FitNet (over 2% mAP). By moving the distillation position to RoIAlign layer, the performance of both methods improve, while WKD-F still performs better than FitNet by a non-trivial margin.

**How spatial size of RoIAlign features affect performance?** In Faster-RCNN, the RoIAlign layer outputs standard 7×7 feature maps. To exploit more spatial information, we let RoIAlign layer output feature maps of higher resolution. Table 12b shows effect of size of feature maps on performance. It can be seen that when the spatial size enlarges mAP increases accordingly, and the mAP tends to saturate if the size is as large as 28×28. The result suggests that larger size of RoIAlign features benefit feature distillation.

| Method | Position of features | mAP | $AP_{50}$ | $AP_{75}$ |
|---|---|---|---|---|
| FitNet | FPN | 34.13 | 54.16 | 36.71 |
| WKD-F (ours) | | **36.40** | **56.52** | **39.64** |
| FitNet‡ | RoIAlign | 36.45 | 56.93 | 39.72 |
| WKD-F (ours) | | **37.21** | **57.32** | **40.15** |

(a) Effect of position of feature distillation. ‡ Reproduced by us.

| Size of feature maps | mAP | $AP_{50}$ | $AP_{75}$ |
|---|---|---|---|
| 7×7 | 36.94 | 57.03 | 40.01 |
| 18×18 | **37.21** | **57.32** | **40.15** |
| 28×28 | 37.15 | 57.23 | 40.11 |

(b) Effect of size of RoIAlign feature maps.

Table 12: Ablation analysis of feature distillation (i.e., WKD-F) for object detection on MS-COCO.

| Faster RCNN-FPN | RN101→RN18 | | |
|---|---|---|---|
| | mAP | $AP_{50}$ | $AP_{75}$ |
| FCFD | 37.37 | 57.60 | 40.34 |
| +WKD-L+WKD-F | **37.66** | **58.01** | **40.66** |
| ReviewKD | 36.75 | 56.72 | 34.00 |
| +WKD-L+WKD-F | **37.50** | **57.79** | **40.38** |

Table 13: Our WKD benefits state-of-the-art KD methods for object detection.

## E.3 Integration with State-of-the-art KD Methods

We assess whether our methodology is complementary to two state-of-the-art methods, i.e., ReviewKD and FCFD. To this end, for ReviewKD, we add the losses of WKD-L and WKD-F into the original loss functions; the weights of our two losses and hyper-parameters are same as those in Section E.1. For FCFD, we replace respectively the original KD loss and feature distillation loss (i.e., FitNet loss) by the losses of WKD-L and WKD-F whose weights are identical to those specified in the main paper. The results are shown in Table 13. We can see that, by integrating our methods, both FCFD and ReviewKD improve by non-trivial margins, which indicates that our methodology is parallel to them and thus can enhance their performance.

## E.4 Implementation Details on BB Regression for KD

Besides common logit distillation and feature distillation, FCFD [8] additionally uses BB regression for knowledge transfer and achieves state-of-the-art performance. Here we also introduce it into our methodology. Specifically, for each proposal from RPN of the student [47], both student and teacher perform BB regression, predicting separately a localization offset vector (LOV) $\mathbf{o}$, from which we obtain predicted bounding box $B$ of the target class. Then the distillation loss is written as

$$\mathcal{L}_{\mathrm{BB}} = \mathcal{L}_2\big(\mathbf{o}^{\mathcal{T}}, \mathbf{o}^{\mathcal{S}}\big) + \xi \mathcal{L}_{\mathrm{DIoU}}\big(B^{\mathcal{T}}, B^{\mathcal{S}}\big), \tag{17}$$

where $\mathcal{L}_2$ indicates the square of Euclidean distance between the student's LOV vector $\mathbf{o}^{\mathcal{S}}$ and the teacher's one $\mathbf{o}^{\mathcal{T}}$. We use the Distance-IoU loss $\mathcal{L}_{\mathrm{DIoU}}$ defined by Zheng et al. [80], which measures the Intersection over Union (IoU) between two bounding-boxes $B^{\mathcal{S}}$ and $B^{\mathcal{T}}$ with a penalty term. The constant $\xi$ is to balance the two losses and is set to 20 throughout.

## F Limitations and Future Research

The cost of our WKD-L is higher than KL-Div based methods due to the regularized linear programming [23]. However, it is affordable as shown in Table 5 and will benefit from advance of faster algorithms for solving WD [45]. Potentially, WKD-L can generalize to be a label smoothing regularization method. Specifically, besides the CE loss, one introduces an additional loss that measures WD rather than KL-Div between the logits and a uniform distribution [81], while computing IRs using embeddings of textual category names as 'prototypes' via ready-made LLMs [82]. Besides BAN based Self-KD in Section 4.5, it is promising to further generalize WKD-L for teacher-free knowledge distillation, e.g., by using customized soft labels [4] and IRs based on weights of the softmax classifier.

Our WKD-F models distribution of features with Gaussians. As deep features of DNNs are generally of high-dimension and small-size, accurate estimation of covariance matrices is difficult [52]. Therefore, exploration of robust and efficient methods for estimating Gaussian may further improve the performance of WKD-F. Besides, the Gaussians may be sub-optimal for modeling feature distributions. To the best of our knowledge, what distributions deep features may exactly follow is an open problem. It is interesting to study other parametric distributions and the corresponding dis-similarities for knowledge distillation.

# G  Broader Impact

We address limitations of Kullback-Leibler Divergence for knowledge distillation (KD), and can obtain stronger, lightweight models suitable for resource-limited devices. Our methodology is very promising, readily applicable to a variety of visual tasks, e.g., image classification and object detection. We hope our work sheds light on importance of WD and inspires future exploration of it in the field of knowledge distillation.

With breakthroughs of large-scale pre-training, multimodal large language models (LLMs) like GPT-4 [83] have excelled in many visual tasks. Our methodology can be potentially applied to transfer knowledge from LLMs to smaller ones for specific visual or language tasks, allowing for improved performance while preserving fast inference cost. Consequently, researchers as well as practitioners can more easily benefit from advanced technologies of LLMs, facilitating their widespread use and broader accessibility.

Currently, the theoretical quest for KD is limited. Due to the black-box nature of the distillation process, the student distilled using our methods may inevitably inherit harmful biases from the teacher [84]. Moreover, the reduction in deployment cost may lead to more potential harms of model abuse. This highlights the necessity for more widespread efforts to regulate the use of artificial intelligence techniques including knowledge distillation.

