# OpenReview forum: "Wasserstein Distance Rivals Kullback-Leibler Divergence for Knowledge Distillation"
_NeurIPS.cc/2024/Conference — NeurIPS 2024 poster_

### Official Review · Reviewer_F2zQ · 2024-07-09

**Soundness:** 3
**Presentation:** 3
**Contribution:** 3
**Rating:** 5
**Confidence:** 4

**Summary:**

This paper proposes a new algorithm for knowledge distillation by replacing KL-divergence loss with Wasserstein distance loss.
The proposed algorithm contains two parts:
(1) Logits distillation with Wasserstein distance loss, implementing with an entropy regularized linear programming;
(2) Assuming the features obey Gaussian distribution, the Wasserstein distance loss can be solved with parametric tricks.

Experiments on ImageNet and CIFAR100 show obvious improvements compared to baselines like KD, DKD, and NKD.

**Strengths:**

(1) The paper is written clearly and is easy to follow.
(2) Sound ablations for combinations of the proposed method and previous ones.

**Weaknesses:**

(1) Eq. (3) and Eq. (5) both have hyperparameter $\lambda$. Does the two lambda have the same value in implementation?
      The ablation for $\lambda$ in Eq. (3) is missing.

(2) There are too many hyper-parameters to be tuned.
       lambda in Eq. (3),
       lambda in Eq. (5),
       k for calculating IR,
       gamma for a trade-off between D_mean and D_cov,
       weight between WKD-L and WKD-F,
       temperature.
       Do all experiments in the paper adopt consistent hyper-parameters?

(3) What's the layer features are used for WKD-F?
      Feature dimensions are usually higher than logits. Moreover, multiple-layer features are often used for feature-based methods.
      Thus, it seems to be wired that WKD-F (207ms) is even faster than the original KD (215ms).

(4) The paper claims that Wasserstein distance rivals Kullback-Leibler divergence for knowledge distillation as indicated by their title.
However, as shown in Table 2(a) and Table 2(b), the KL-div method achieves 71.96 in the setting of separating target and non-target.
The WD-based method just achieves comparable results with the Polynomial kernel, class centroid, or classifier weight for calculating IR.

It seems that the IR is more important than the form of WD formulation.

**Questions:**

(1) Ablation for $\lambda$ in Eq. (3).
(2) Explanation and analysis of sensitivity of all hyper-parameters.
(3) How to calculate $\mu^T$ and $\mu^S$ in your implementation for Eq. (9)?
(4) Does the proposed method work well in the setting of self-KD where the student and teacher share the same architecture?

**Limitations:**

Limitations are discussed in the paper

---

> ### Author Rebuttal · Authors · 2024-08-07
>
> Dear Reviewer F2zQ,
>
> Thank you for reviewing our paper and for providing constructive and insightful comments. We appreciate your acknowledgement of good soundness, presentation \& contribution of our paper. We hope our detailed responses can address your concerns, and could you please consider increasing your rating accordingly.
>
> > ### Eq. (3) and Eq. (5) both have hyperparameter $\lambda$. Does the two lambda have the same value in implementation? The ablation for the hyperparameter in Eq. (3) is missing.
>
> We apologize for our typo that leads to confusion.
>
> The **hyper-parameter (HP) in Eq. (3) is a typo and will be denoted by $\eta$**, indicating the regularizing constant of entropic term, while  $\lambda$ in Eq. (5) indicates the weight of WD loss; they have different values. We ablate $\eta$ in Setting (a) and results (Acc, \%) are shown below along with Figure R1 in PDF for Global Rebuttal. We see across a decently large range the performance changes smoothly, particularly for Top-5 Acc. Notably, **$\eta$ in Eq. (3) is always set to 0.05 across the paper (Line 231)** in either classification on ImageNet \& CIFAR100 or object detection on MS-COCO.
>
> Method|$\mid$|$\quad \eta$  |$\mid$|Top-1|$\mid$|Top-5|
> |:-:|:-:|:-:|:-:|:-:|:-:|:-:|
> |KD|$\mid$|NA|$\mid$|71.03|$\mid$|90.05|
> ||
> |WKD-L|$\mid$|1.0e-2|$\mid$|71.65|$\mid$|90.21|
> |--|$\mid$|2.0e-2|$\mid$|72.15|$\mid$|90.70|
> |--|$\mid$|5.0e-2|$\mid$|**72.49**|$\mid$|**90.75**|
> |--|$\mid$|1.0e-1|$\mid$|72.25|$\mid$|90.74|
> |--|$\mid$|2.0e-1|$\mid$|72.09|$\mid$|90.53|
> ||
>
> > ### Explanation and analysis of sensitivity of all hyper-parameters.
>
> Above all, kindly note **as logit-based WKD-L and feature-based WKD-F are two distinct methods, their hyper-parameters (HPs) should be analyzed separately.**
>
> WKD-L is to address the downside of lack of cross-category comparison in classical KL-Div. **We ablate its HPs containing temperature, sharpening parameter and weight in Section B.1 (Lines 636--649 and Figure 5),** where we see that they are insensitive in decently large ranges. $\eta$ in Eq. (3) is always fixed across the paper.
>
> WKD-F aims to address the problem of KL-Div that cannot exploit geometry of data manifold. **We ablate its HPs (i.e., mean-cov ratio and the weight) in Section B.1 (Lines 650--659 and Figure 6),** and observe that they are invulnerable to large variation of values.
>
> Further, combining WKD-L and WKD-F leads to further improvement; in this case, we simply add their losses **where all HPs are unchanged.**
>
> Finally, for classification on ImageNet a summary of HPs is given in Section B.2; on CIFAR100, for a fair comparison, we follow DKD [3], CAT [51], ReviewKD [29], FCFD [8] and WTTM [5] *for KD within CNNs,* and follow OFA [62] *for KD across CNNs and Transformers,*  tuning HPs separately for different settings (Section C). For object detection, the framework of Faster-RCNN is quite different from that of classification, and how HPs are turned are given in Section E.
>
> > ### How to calculate $\mu^S$ and $\mu^T$ in your implementation for Eq. (9)?
>
> On Lines 602--610 we explain concretely how to compute the means. For teacher, we calculate directly $\mu^T$ using features output by **single stage of Conv5_x with 1x1 spatial grid**; for student we use the corresponding features after the projection layer  for calculating $\mu^S$.
>
> > ### Does the proposed method work well in the setting of self-KD where the student and teacher share the same architecture?
>
> Thanks for the suggestion.
>
> We implement self-KD in the framework of Born-again network (BAN) (Furlanello et al., ICML’18). Specifically, we first train a network model $S_{0}$ with ground truths. With $S_{0}$ as the teacher, we use WKD-L and distill a new model $S_{1}$ with same architecture. For simplicity's sake, we do not perform the 2nd-generation distillation, i.e., distilling a new model $S_{2}$ with $S_{1}$ as the teacher, etc.
>
> We experiment with ResNet18 and the HPs are same as those in Setting (a). For logit-distillation, state-of-the-art USKD [4] is slightly better than BAN (KL-Div), both improves the regular training without self-KD; our BAN (WKD-L) performs much better than USKD, suggesting superiority of WKD-L in self-KD setting.
>
> |$\ \ $Method|\||Self-KD |\||Dis-similarity|\||Top-1|
> |:-:|-|:-:|-|:-:|-|:-:|
> |Regular|\||$\times$|\||NA|\||69.75|
> ||
> |USKD|\||$\checkmark$|\||KL-Div|\||70.75|
> |BAN|\||$\checkmark$|\||KL-Div|\||70.50|
> |BAN (WKD-L)|\||$\checkmark$|\||WD|\||**71.35**|
> ||
>
>
>
> > ### What's the layer features are used for WKD-F? It seems to be wired that WKD-F (207ms) is even faster than the original KD (215ms).
>
> For WKD-F, we use the **features of single stage of Conv5\_x,** i.e., 49 features of 512-dimension that is lower than logits (i.e., 1000). Our ablation in Table 3(c) shows that single stage with 1x1 grid is the best option. Kindly note that for WKD-F, we only need to compute the mean and variances for Gaussian (Diag) that are very efficient. To resolve your concern, we also measure latencies (ms) in Setting (b), where we have 49 features of 2048-dimension for distribution matching. Compared to Setting (a), the latencies of both methods increase while that of WKD-F is slightly higher.
>
> |Method|$\mid$|Strategy|$\mid$|Dimension|$\mid$|Latency|
> |:-:|:-:|:-:|:-:|:-:|:-:|:-:|
> |Setting (a)
> |KD|$\mid$|Logit|$\mid$|1000|$\mid$|215|
> |WKD-F|$\mid$|Feature|$\mid$|512|$\mid$|207|
> ||
> |Setting (b)
> |KD|$\mid$|Logit|$\mid$|1000|$\mid$|268|
> |WKD-F|$\mid$|Feature|$\mid$|2048|$\mid$|276|
> ||
>
>
>
>
> > ### It seems that the IR is more important than the form of WD formulation.
>
> Thanks for the concern.
>
> Kindly note **IR is a key ingredient of our WD formulation,** without which IRs cannot play the roles, e.g., IRs cannot be used in KL-Div based knowledge distillation.
>
> It is worth noting that the results in Table 2(b) show that WD formulation with each IR performs better than strongest KL-Div variant [4] (71.96), suggesting it is superior to KL-Div formulation.

---

> > ### Comment · Reviewer_F2zQ · 2024-08-14
> >
> > Thanks for the responses from the authors.
> >
> > My concerns have been addressed.

---

> > > ### Author Response · Authors · 2024-08-14
> > >
> > > Dear Reviewer F2zQ,
> > >
> > > Thank you for your positive feedback. We are delighted to hear that our rebuttal has effectively addressed your concerns. The responses to your constructive comments will be incorporated into the revised paper. We would greatly appreciate it if you could consider reflecting this in your updated score.

---

### Official Review · Reviewer_5GY4 · 2024-07-12

**Soundness:** 3
**Presentation:** 3
**Contribution:** 4
**Rating:** 7
**Confidence:** 3

**Summary:**

This paper proposes a Wasserstein distance based knowledge distillation method for both the logit distillation and feature distillation settings. The logit based version uses discrete WD to model the discrepancy beteween the prediction probabilites of student and teacher networks. It further uses the separation of target probability to improve performance. The feature based version minimizes the continuous WD between the patch features of an image from the student and teacher networks, under the assumption that they form Gaussian distributions. The covariance term is simplifed to its diagnoals to further improve performance. Comapred to recent baslines, the proposed method shows superior performance on ImageNet classification and COCO object detection.

**Strengths:**

1. Strong performance compared to recent work.
2. The presentation is easy to follow and well structured. Related works are introduced to give good contexts.
3. Extensive comparison between baselines, detailed abalation study, and has extra experiments on distillation across CNNs and transformers in appendix.

**Weaknesses:**

One of the motivation for the WKD-L is the cross-category comparison, however, it is not clear to me why the "cross-category" comparison is attrubted as the source of improvement. Firstly, there is implicit cross-category comparision for KL based methods (line 117), so it is not a differentiator. Secondly, without the story of cross-category comparison, the WDK-F also shows improvement.

**Questions:**

1. Could you explain the statement at line 120 "this implicit effect is insignificant"?
2. Line 38. "... features of an image are ... small size". What does the "small size" of an image referes to?
3. Are the student networks randomly initialized or initialized from a trained weights for baseline methods (line 239)?
4. Line 172 "... partition the feature maps into a kxk spatial grid", and line 289 "use ... 1x1 grid for classification". There is no grid setting for detection in the main text and can mislead readers to think grid is not used at all.

**Limitations:**

The authors discussed the limitations.

---

> ### Author Rebuttal · Authors · 2024-08-07
>
> Dear Reviewer 5GY4,
>
> Thank you for reviewing our paper and for providing constructive and insightful comments. Particularly, we appreciate your acknowledgement that our paper has excellent contribution, strong performance and good presentation. We hope our detailed responses can address your concerns, and could you please consider increasing your rating accordingly.
>
>
> > ### Could you explain the statement at line 120 "this implicit effect is insignificant"? it is not clear to me why the "cross-category" comparison is attributed as the source of improvement.
>
> Thanks for your concerns.
>
> First, please allow us to explain what "implicit effect" exactly means in classical KD that is a logit-based distillation method. While comparing WKD-L (Figure 1b (left)) to classical KD (Figure 2), we can see that KL-Div only performs category-to-category comparison, **but lack of mechanism to *explicitly* exploit cross-category comparison via pair-wise IRs in Eq. (1),** which contain rich knowledge among categories. Nevertheless, for KL-Div based KD, through backpropagation the probability of one category affects probabilities of the other categories due to softmax function; in this sense, we say that the cross-category effect in KL-Div is implicit.
>
> By stating "this implicit effect is insignificant", we mean that **gains brought by explicit cross-category comparison in WD is more significant than that brought by the implicit influence in KL-Div, as experimentally shown in Table 2(a).** Specifically, WD vs. KL-Div in Top-1 Acc (%) is 72.04 vs. 71.03 without (w/o) target separation and is 72.49 vs. 71.96 with (w/) target separation. As such, we attribute the improvement to cross-category comparison inherent in WD.
>
> Additionally, kindly note that WKD-F belongs to the type of feature-based distillation methods, quite different from the type of logit-based ones, such as WKD-L and classical KD & its variants. Therefore, the effectiveness of WKD-F cannot suggest that the improvement in WKD-F is not attributed to cross-category comparison.
>
> We are sorry for the possible ambiguity and we’ll make further clarification in the modified paper.
>
> > ### What does the "small size" of features of an image refers to (Line 38)?
>
> Thanks for the concern.
>
> **Here "small size" indicates that for an input image the number of convolutional features output by a CNN is small.** For example, for ResNet50 with a standard 224x224 input image, the commonly used feature maps of stage 5 (Conv\_5x) contain only 49 (7$\times$7) features, which however are of high-dimension (i.e. 2048).
>
> The high-dimensional features of small size not only makes non-parametric density estimation (e.g., histogram) infeasible, but also leads to non-overlapping discrete distributions, both bringing  difficulty for KL-Div. One may turn to parametric distributions (e.g., Gaussian); however, KL-Div is limited as it is not a true metric and is unaware of geometry of the underlying manifold. **This downside of KL-Div motivates our 2nd contribution, i.e., WKD-F that uses Wasserstein distance as dis-similarity between Gaussians.**
>
> > ### Are the student networks randomly initialized or initialized from a trained weights for baseline methods (line 239)?
>
> Thanks for the concern.
>
> For fair comparison with previous arts, we use the framework of Faster-RCNN for object detection. **For the student models, the backbones are initialized with pre-trained weights on ImageNet, while Region Proposal Network (RPN), Feature Pyramid Network (FPN), and detection head consisting of a classification branch and a localization branch are all initialized randomly.**  Note that this is a standard practice widely adopted by knowledge distillation methods for object detection, as well as by all general object detection methods.
>
> > ### There is no grid setting for detection in the main text and can mislead readers to think grid is not used at all.
>
> Thanks for your suggestion.
>
> We are sorry for the possible ambiguity. As described in Section E.1 in Appendix, for object detection, we use a 4$\times$4 spatial grid for computing Gaussian. We’ll make this clear in the main text.

---

> > ### Comment · Reviewer_5GY4 · 2024-08-13
> >
> > Thanks for the rebuttal. It clarifies my questions and I will maintain the score.

---

> > > ### Author Response · Authors · 2024-08-13
> > >
> > > Dear Reviewer 5GY4,
> > >
> > > We are pleased to learn that we have effectively addressed your concerns. We appreciate your very positive comments on the soundness, presentation and contribution of our paper.

---

### Official Review · Reviewer_Zit9 · 2024-07-12

**Soundness:** 3
**Presentation:** 2
**Contribution:** 3
**Rating:** 6
**Confidence:** 4

**Summary:**

The paper introduces a novel methodology for knowledge distillation using Wasserstein Distance (WD) instead of the traditional Kullback-Leibler Divergence (KL-Div). The proposed methods include a logit distillation approach (WKD-L) that leverages cross-category comparisons and a feature distillation method (WKD-F) that models feature distributions parametrically. The authors demonstrate that their methods outperform strong KL-Div variants on image classification and object detection tasks.

**Strengths:**

* The introduction of WD in knowledge distillation provides a fresh perspective and addresses the limitations of KL-Div, particularly in terms of cross-category comparisons and handling non-overlapping distributions.
* The use of parametric methods for feature distribution modeling, specifically Gaussian distributions, is innovative and effectively leverages the geometric structure of the data.

**Weaknesses:**

* The paper suffers from several writing issues, including grammatical errors and unclear explanations, making it difficult to follow the arguments and methodology at times.
* Some of the assumptions made for the application of WD, particularly the choice of Gaussian distributions for feature modeling, may not hold universally. Further justification or exploration of alternative parametric methods would strengthen the work.
* The computational complexity of implementing WD-based methods, especially in large-scale scenarios, is not adequately addressed. A comparison of computational costs between WD and KL-Div would be beneficial.

**Questions:**

* Why did you choose Gaussian distributions for feature modeling in the context of WD? Are there other parametric methods that you considered, and how would they compare in terms of performance and feasibility?
* How does the computational complexity of your proposed WD-based methods compare to traditional KL-Div based methods? Can you provide a detailed analysis or empirical comparison of the computational costs involved?

**Limitations:**

* please check the questions and weaknesses.

---

> ### Author Rebuttal · Authors · 2024-08-07
>
> Dear Reviewer Zit9,
>
> Thank you for reviewing our paper and for providing constructive and insightful comments. We are grateful for your positive comments on novelty and contribution of WD based methods, such as "a fresh perspective and addresses the limitations of KL-Div" and "is innovative and effectively leverages the geometric structure of data". We hope our detailed responses can address your concerns, and could you please consider increasing your rating accordingly.
>
> > ### Why did you choose Gaussian distributions for feature modeling in the context of WD? Are there other parametric methods that you considered, and how would they compare in terms of performance and feasibility?
>
> Thanks for your concerns.
>
> Kindly note that, **we have discussed why we choose Gaussians on Lines 63--68 in Section 1 and Lines 154--157 in Section 2.2.** Specifically, Gaussian is of maximal entropy for given 1st-moment (mean) and 2nd-moments (covariance) estimated from random samples (features) among the family of parametric distributions. Namely, it has maximum uncertainty with the least prior and thus is more general than other candidates. In addition, WD between Gaussians is a Riemannian metric with closed-form solution; in contrast, WDs between other parametric distributions are open problems, as far as we know.
>
> **We have compared to other parametric methods in Table 3 (a) and Lines 267--276**, including *separate spatial 1st- or 2nd-moments [35]* as well as *separate channel 1st- or  2nd-moments [6]*. We note that both [35] and [6] fail to consider the geometry of the underlying manifold; kindly refer to Lines 205--218 and Table 1 for our discussion. The experiments show that our method outperform them by large margins with insignificant increase of computation (See Table 5).
>
> **Following your suggestion, we consider additional parametric distributions** including Laplace and exponential distributions. We use univariate Laplace or exponential distribution for modeling each component of features. For them KL-Div can be computed in closed-form [Ref1] but WD is an unsolved problem. For Gaussian (Diag) we use univariate Gaussian for each feature component. We experiment with the same setting as in Table 3 (a) and the comparison is shown in the table below. We firstly note that, *with KL-Div*, Gaussian is better than both Laplace and exponential functions, which suggests Gaussian is a better option among the competing parametric distributions. Furthermore, Gaussian with WD outperforms Gaussian with KL-Div, which we attribute to the fact that WD can effectively exploit geometry of the underlying manifold but KL-Div cannot.
>
> |Distribution |$\mid$|Dis-similarity|$\mid$|Top-1|
> |:-:|:-:|:-:|:-:|:-:|
> |Laplace|$\mid$|KL-Div|$\mid$|71.38|
> |Exponential|$\mid$|KL-Div|$\mid$|70.14|
> |Gaussian (Diag)|$\mid$|KL-Div|$\mid$|71.75|
> ||
> |Gaussian (Diag)|$\mid$|WD|$\mid$|**72.50**|
> ||
>
> [Ref1] M. Gil. On Rényi divergence measures for continuous alphabet sources. Master's thesis
> , 2011.
>
> > ### How does the computational complexity of your proposed WD-based methods compare to traditional KL-Div based methods? Can you provide a detailed analysis or empirical comparison of the computational costs involved?
>
> Thanks for the comments.
>
> Kindly note that, **we have provided an empirical comparison of the computational cost in Table 5 and Lines 306--317.**  Compared to traditional KL-Div, our WKD-L, the logit-based distillation method, has somewhat higher cost due to the regularized linear programming; this limitation is discussed in Section F. Meanwhile, our WKD-F, the feature-based distillation method, has comparable cost with traditional KL-Div.
>
> **Following your suggestion, we further conduct theoretical analysis.** The logit-based WKD-L is formulated as an entropy regularized linear programming, which can be solved by fast Sinkhorn algorithm [23]. Let $n$ be the dimension of the predicted logits, the computational complexity of WKD-L can be written as is $O(Dn^2 \log n) $ [Ref2]. Here  $D=\|\|C\|\|\_{\infty}^{3}\epsilon$ is a constant, where $\|\|C\|\|\_{\infty}$ indicates the infinity norm of the transportation cost matrix $C = [c_{ij}]$, and $\epsilon>0$ indicates a prescribed error. In contrast, the computational complexity of KL-Div is $O(n)$. Despite its high complexity, WKD-L can be computed efficiently as Sinkhorn algorithm is highly suitable for parallel computation on GPU [23]. For feature-based WKD-F, the dominant cost is due to computation of means and variances. Give a set of $m$ features $\mathbf{f}_{i}$ of $l$-dimension, the means can be computed by global average pooling (GAP) that takes $O(ml)$ time; the complexity of variances is also $O(ml)$ which can be implemented by element-wise square operations followed by a GAP.
>
> [Ref2] J. Altschuler, et al. Near-linear time approximation algorithms for optimal transport via sinkhorn iteration. In NeurIPS, 2017.
>
>
> > ### Some of the assumptions made for the application of WD, particularly the choice of Gaussian distributions for feature modeling, may not hold universally.
>
> Thanks for the comment.
>
> We agree that the Gaussian distributions may not hold universally. **Kindly note that this limitation has been discussed in Section F.** Specifically, what distributions deep features may exactly follow is an open problem, which is rarely studied in deep learning to our best knowledge. As such, Gaussian may be sub-optimal for modeling distributions of DNN features, and it is interesting to explore other parametric distributions.  In addition, the Wasserstein distances between parametric distributions other than Gaussian are open questions in probability and statistics and worth future research.
>
> > ### The paper suffers from several writing issues, including grammatical errors and unclear explanations, making it difficult to follow the arguments and methodology at times.
>
> Thanks for your comment. We’ll further polish our manuscript, carefully addressing writing issues.

---

> > ### Comment · Reviewer_Zit9 · 2024-08-13
> > **Response by Reviewer Zit9**
> >
> > I have carefully reviewed the feedback from other reviewers, considered the author’s rebuttal, and global responses, followed the ensuing discussion, and read the paper again . I appreciate the authors' thorough responses, particularly their clarification on W2  and new experimental results during the rebuttal period as well as Q2. Itherefore I will raise my score slightly from 5 to 6. Good luck!

---

> > > ### Author Response · Authors · 2024-08-13
> > >
> > > Dear Reviewer Zit9,
> > >
> > > Thank you for your positive feedback on our rebuttal and for raising your score. We are pleased to hear that your concerns have been satisfactorily addressed. The responses to your constructive comments will be incorporated into the revised paper.

---

### Official Review · Reviewer_ScyW · 2024-07-17

**Soundness:** 3
**Presentation:** 3
**Contribution:** 3
**Rating:** 5
**Confidence:** 4

**Summary:**

The paper proposes the utilization of Wasserstein Distance based distillation as opposed to KLD, as is common in practice. This is because the latter does not facilitate cross-category comparisons. Both logit and feature based variants have been proposed. The comparisons have been shown for both classification as well as detection.

**Strengths:**

Though straightforward look at the empirical gains are not massive, the method is theoretically sound, and the paper is easy to understand. The experiments are decently presented in my opinion. The paper provides a fresh perspective of WD in distillation.

**Weaknesses:**

I would have been happier to see if changing the divergence led to more significant boosts, if at all possible (basis the premise of providing cross category comparisons).

**Questions:**

None

**Limitations:**

The cost expensiveness has been mentioned.

---

> ### Author Rebuttal · Authors · 2024-08-07
>
> Dear Reviewer ScyW,
>
> Thank you for reviewing our paper and for providing constructive and insightful comments. Particularly, we appreciate your positive comments on novelty ("a fresh perspective of WD in distillation"), on soundness ("theoretically sound"), and presentation \& contribution ("good"). We hope our detailed responses can address your concern, and could you please consider increasing your rating accordingly.
>
> > ### I would have been happier to see if changing the divergence led to more significant boosts, if at all possible (basis the premise of providing cross category comparisons).
>
> Thanks for your concern.
>
> Our WKD provides a novel viewpoint of WD in knowledge distillation (KD), which have shown very competitive performance, compared to pre-dominant KL-Div based knowledge distillation methods. Considering that WD is overlooked and rarely studied in KD, we believe that **the promising results of our work can inspire great interest and the follow-up works have potentials to yield larger performance boosts over the KL-Div based ones.**
>
> It is worth mentioning that **the performance improvement of WKD is non-trivial in the *commonly used settings*** which has long been studied. For example, in the setting (a) on ImageNet, for logit distillation WTTM (SOTA of 2024) improves NKD (SOTA of 2023) by 0.23 percentage points (PPs) while the latter surpasses DKD (SOTA of 2022) by 0.26 PPs; in contrast, WKD-L outperforms WTTM by 0.30 PPs; for feature distillation our WKD-F surpasses the SOTA method of ReviewKD by 0.89 PPs. Finally, we note that **our WKD considerably outperforms the competitors *in the new setting of across CNN and vision transformers***, as shown in Table 9(a) in Section C.2 of Appendix.

---

### Author Rebuttal · Authors · 2024-08-07

Dear Reviewers, Area Chairs and Program Chairs,

We thank all reviewers again for their thoughtful and constructive comments, which are helpful in improving the quality of our paper. After carefully reading all comments and questions, we conducted additional experiments and discussions to address the reviewers' concerns. In the following, we provide a summary of the reviewers' comments and our responses.

**Firstly**, we are pleased to observe that majority of reviewers have highly recognized the innovativeness, clarity of presentation, experiments and performance of our work, initially recommending **1 accept** and **3 borderline accept**. Specifically, we are grateful for the following positive comments about our work:

* The paper provides **a fresh perspective** of WD in distillation (Reviewers ScyW and Zit9) and **a new algorithm** for knowledge distillation (Reviewer F2zQ); the use of parametric methods for feature distribution modeling, specifically Gaussian distributions, **is innovative.** (Reviewer Zit9).
* The proposed methods exhibit **strong performance** compared to recent work (Reviewer 5GY4), and show **obvious improvements** compared to baselines (Reviewer F2zQ). The experiments conduct **extensive comparison** (Reviewer 5GY4), **are decently presented** (Reviewer ScyW)  and provide **sound ablations** (Reviewer F2zQ). The paper provides **extra experiments on distillation *across CNNs and transformers*** in appendix (Reviewer 5GY4).
* The paper is **easy to understand** (Reviewer ScyW), **easy to follow, written clearly** (Reviewer F2zQ) and **well structured** (Reviewer 5GY4). Additionally, the related works are introduced to **give good contexts** (Reviewer 5GY4).

---

**Secondly**, we provide point-by-point responses to all four reviewers, striving to address their concerns and questions. We will carefully revise our manuscript according to these discussions. Here, we list major experiments and discussions that have been added in responses.

> ### 1. Compared to other parametric methods for feature distribution modeling.

**Following the suggestion of Reviewer Zit9**, we compare to additional parametric distributions including Laplace and exponential distributions. **With KL-Div**, Gaussian distribution is a better option in capturing the feature distribution and knowledge transferring. **By leveraging the Wasserstein distance (WD)**, Gaussian distribution achieved further improvements in performance. For detailed results and discussions, please refer to our response to Reviewer Zit9 and Table R1 in the attached PDF. It is worth mentioning that **we have already compared to parametric methods including 1st- or 2nd-moments (spatial and channel)** in the paper (Table 3 (a) and Lines 267--276).

> ### 2. Additional experiments on self-KD settings.

**Following the suggestion of Reviewer F2zQ**, we implement self-KD in the framework of Born-again network (BAN) (Furlanello et al., ICML’18). Our **WKD-L performs much better than classical KD and the state-of-the-art method USKD**, suggesting the effectiveness of WKD in self-KD settings. The detailed results can be found in our response to Reviewer F2zQ and Table R2 in the attached PDF.

> ### 3. Ablation study on the regularizing constant of entropic term in Eq. (3).

**Following the suggestion of reviewer F2zQ**, we have conducted an ablation study on the regularizing constant of the entropic term $\eta$ in Eq. (3). Across a decently large range, the performance changes smoothly, indicating that our method is not sensitive to $\eta$. **Note that $\eta$ is always fixed across the paper.** The detailed results are provided in our response to Reviewer F2zQ and Figure R1 in the attached PDF.

> ### 4. Evaluation of training latency for Setting (b).

We have measured training latencies (ms) for Setting(a) in Table 5, where WKD-F is slightly faster than classical KD. **To resolve the concern of reviewer F2zQ**, we further measure the latencies for Setting (b), where the latencies of both methods increased while that of WKD-F is slightly higher. The detailed results are shown in response to Reviewer F2zQ and Table R3 in the attached PDF.

---

**Finally**, based on the constructive comments and suggestions from all reviewers, we will carefully revise our manuscript. We believe we can effectively address these concerns, and we kindly ask you to consider increasing your scores. We are looking forward to your positive feedback on our rebuttals, and please feel free to reach out if you have any additional questions.

---

### Comment · Area_Chair_HLsh · 2024-08-12

Dear reviewers, could you share your feedback to the authors after reading their response? Does the authors' response address your concerns well?

---

### Decision · Program_Chairs · 2024-09-25

**Decision:**

Accept (poster)

**Comment:**

The paper provides an interesting perspective of WD in distillation and a new algorithm for knowledge distillation. After initial rating, the authors provide a nice rebuttal which addressed all the issues mentioned by the reviewers. Now, all the reviewers though there is no further issues with this paper and the AC also learns toward an acceptance for this submission.